# COMMUNITY-AWARE HARD SUBGRAPH MINING FOR OUT-OF-DISTRIBUTION GENERALIZATION

## ABSTRACT

Graph Neural Networks (GNNs) are widely used for node classification tasks but often struggle to generalize when training and test nodes follow different distributions, limiting their real-world applicability. Recent approaches based on invariant learning attempt to address this issue but rely on impractical predefined environment labels or low-quality synthetic environments, and their strict invariance assumptions often fail under complex community-level variations. In this work, we propose **CHASM** (*Community-Aware Hard Subgraph Mining*), a novel framework for OOD generalization on graphs that explicitly leverages latent community heterogeneity. CHASM adversarially mines the hardest subgraphs via a learnable mask model, imposes community-aware regularization to enforce structural coherence, and applies adaptive subgraph augmentation to enhance robustness. A stability-driven learner is then optimized against these hardest cases, yielding a principled and effective solution to community-level shifts. Extensive experiments under covariate and concept shifts demonstrate that CHASM consistently outperforms state-of-the-art baselines, while theoretical analysis provides further justification of its robustness.

## 1 INTRODUCTION

Graph Neural Networks (GNNs) (Kipf & Welling, 2017; Veličković et al., 2018; Wu et al., 2019; Hamilton et al., 2017) have demonstrated remarkable success in learning node representations from graph-structured data, and have been widely applied to node classification tasks such as recommendation systems (Jiang et al., 2023) and anomaly detection (Tang et al., 2022) in social networks. Despite their effectiveness, most GNN-based methods are built upon the assumption that training and test nodes are drawn from the same distribution, implying that both node features and graph structures follow consistent patterns across different phases of learning. However, this assumption rarely holds in practice. Real-world graphs often exhibit strong heterogeneity across latent communities: for example, user groups in social networks may display vastly different interaction patterns and attribute distributions, while research communities in citation networks may evolve their structural and topical focus over time (Newman, 2001; Ugander et al., 2011; Mislove et al., 2007). Such discrepancies between communities can create substantial distribution gaps between the training and testing phases. Consequently, the training distribution may fail to cover all variations encountered during deployment, leading to severe performance degradation. This phenomenon, where a discrepancy arises between the training and testing distributions of nodes and structures, is referred to as the out-of-distribution (OOD) problem. GNNs trained with empirical risk minimization (ERM) suffer significant performance drops under OOD scenarios, motivating the need for principled approaches that explicitly handle distribution shifts in graph data.

To address the OOD problem in graphs, existing studies have explored several strategies such as data augmentation (Wang et al., 2021) and invariant learning (Liu et al., 2023; Wang et al., 2024; Li et al., 2025). Data augmentation methods expose models to perturbed features or structures in order to reduce overfitting to spurious correlations. While effective to some extent, such augmentations rely on manually designed perturbation rules and cannot guarantee that the synthesized distributions accurately reflect real unseen environments. Invariant learning-based approaches aim to identify causal features that remain predictive across different environments, thereby mitigating performance degradation under distribution shifts. Although this paradigm has recently become a mainstream solution for OOD generalization, its applicability to graphs is limited. First, environment labels are

often unavailable in real-world graph data, making it difficult to explicitly construct environment-specific risks. Second, environments synthesized via data augmentation are often heuristic and fail to faithfully capture the true heterogeneity of graph data. For instance, common augmentation strategies such as random edge perturbation or feature masking mainly introduce local noise, but they do not reflect the community-level variations that naturally arise in real graphs. Third, enforcing strict invariance across all environments can be overly restrictive and lacks theoretical guarantees in scenarios where community-level variations dominate. This raises a natural question:

*Can we leverage the latent community heterogeneity in graphs to improve OOD generalization without relying on pre-defined environment labels?*

Different from invariant learning methods that rely on artificially constructed environments through data augmentation, we directly exploit the intrinsic heterogeneity of graph data. Graphs in real-world applications, such as social networks, naturally consist of latent communities whose structural and feature distributions may differ significantly. Such community-level heterogeneity naturally gives rise to *hard subgraphs*, where prediction mechanisms deviate the most from the global one. Rather than synthesizing virtual environments that may not faithfully capture the data-generating process, we explicitly mine these hard subgraphs and measure the discrepancies between their prediction rules and the global predictor. To achieve this, we adopt an adversarial learning perspective: the model adaptively extracts the hardest subgraphs that maximize prediction discrepancies, while simultaneously learning to close these gaps for improved robustness.

Building on this insight, we introduce **CHASM** (*Community-Aware **H**ard **S**ubgraph **M**ining*), a novel framework for OOD generalization on graphs. CHASM automatically identifies challenging subgraphs exhibiting strong community-level discrepancies, assigns adaptive weights to vulnerable nodes, and incorporates community-aware regularization to preserve structural coherence. Furthermore, to enhance robustness, CHASM introduces an adaptive augmentation strategy that perturbs features and edges based on distributional differences between the hardest subgraphs and the overall training graph, thereby generating task-relevant diversity and improving resilience to community-level shifts. Extensive experiments on benchmark OOD node classification tasks demonstrate that CHASM consistently outperforms state-of-the-art methods, validating the effectiveness of leveraging community heterogeneity through hard subgraph mining for robust graph learning.

## 2 PRELIMINARIES

**Notations.** A graph is denoted as $G = (V, E)$, where $V = \{v_1, \ldots, v_N\}$ represents the set of $N$ nodes and $E \subseteq V \times V$ denotes the edge set. Each node $v_i \in V$ is associated with a $d$-dimensional feature vector $x_i \in \mathbb{R}^d$, and the feature matrix of all nodes is denoted as $X = [x_1; \ldots; x_N] \in \mathbb{R}^{N \times d}$. The adjacency structure of the graph is represented by the matrix $A \in \{0, 1\}^{N \times N}$, where $A_{ij} = 1$ if $(i, j) \in E$ and 0 otherwise. $\mathcal{N}(i)$ denotes the neighborhood set of node $v_i$, and $Y = \{y_v \mid v \in V\}$ to denote the ground-truth labels of nodes.

### 2.1 INVARIANT LEARNING

Following the typical OOD setting (Liu et al., 2021), we assume a graph dataset under distribution shifts is collected from multiple latent environments $e \in \text{supp}(\mathcal{E}_{tr})$, where environment labels are typically unavailable. Let $\mathcal{E}_{tr}$ denote the random variable of training environment indices, and $\mathbb{P}^e$ the distribution of nodes and labels in environment $e$.

**Problem 2.1** (Node OOD Generalization under Invariant Learning). *The goal is to learn a GNN predictor $h_\psi(\cdot) : \mathcal{X}, \mathcal{A} \to \mathcal{Y}$ that generalizes across environments:*

$$h^* = \arg\min_h \max_{e \in \text{supp}(\mathcal{E})} \mathbb{E}[l(h(X, A), Y) \mid e], \tag{1}$$

*where $\mathbb{E}[l(h(X, A), Y) \mid e] = \mathbb{E}^e[l(h(X^e, A^e), Y^e)]$ is the expected risk on environment $e$, and $l(\cdot, \cdot)$ is the loss function.*

For unseen environments $e \in \text{supp}(\mathcal{E}) \backslash \text{supp}(\mathcal{E}_{tr})$, the joint distribution $\mathbb{P}^e(X, Y)$ may differ significantly from training environments. Thus, directly solving Equation 1 is challenging without prior

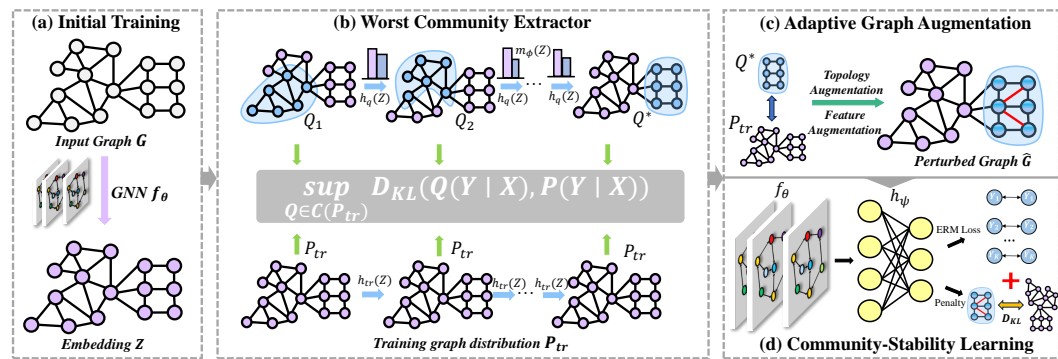

Figure 1: Illustration of the proposed **CHASM** framework, where the adversary mines structurally coherent hard subgraphs, applies adaptive perturbations, and the learner is trained to minimize risks under these challenging conditions to achieve robust community-level OOD generalization.

knowledge or assumptions. This assumption requires the predictive mechanism to be exactly identical across environments, which is often too restrictive in graph learning, since environment labels are typically unavailable and real-world graphs exhibit intrinsic heterogeneity.

## 2.2 LATENT COMMUNITY HETEROGENEITY

**Problem 2.2** (OOD Generalization under Latent Community Heterogeneity). *Given a graph $G = (X, A, Y)$ with node features $X$ and node labels $Y$, where the training nodes $(X, Y) \sim \mathbb{P}_{tr}(X, Y)$ are collected from multiple latent communities without explicit community labels, the problem is to learn node classification models with good generalization performance on test data $(X, Y) \sim \mathbb{P}_{te}(X, Y)$.*

Here the *latent community heterogeneity* refers to the underlying community structures in the graph, where different communities may induce distinct local distributions. The learnability of traditional node classification relies on the i.i.d. assumption that the training and testing distributions are identical. However, in Problem 2.2, where the testing distribution is agnostic and can significantly differ due to unseen or shifted communities, even the basic learnability guarantee may not hold in general.

## 3 METHODOLOGY

### 3.1 OVERALL FRAMEWORK

**Overall Optimization Objective** The goal of **CHASM** (*Community-Aware Hard Subgraph Mining*) is to enhance the robustness of node classification models under latent distribution shifts by explicitly accounting for latent community heterogeneity in graphs. Given a training graph $G = (X, A, Y)$, we formulate the following optimization objective:

$$\min_{\psi, \theta} \underbrace{\mathbb{E}_{P_{tr}}\big[\ell(h_\psi(f_\theta(X, A)), Y)\big]}_{\text{ERM loss}} + \lambda \cdot \underbrace{\sup_{Q \in \mathcal{C}(P_{tr})} D_{\text{KL}}\big(Q(Y|f_\theta(X, A)), \, P_{tr}(Y|f_\theta(X, A))\big)}_{\text{Comunity-aware regularizer}}, \quad (2)$$

where $f_\theta$ is the GNN encoder parameterized by $\theta$, and $h_\psi$ is the classifier parameterized by $\psi$. $\ell(\cdot)$ is the cross-entropy classification loss, and the second term penalizes the maximal KL divergence between the conditional distribution of a community-specific subgraph $Q$ and that of the full training distribution $P_{tr}$. The hyperparameter $\lambda$ controls the trade-off of this regularization term. Here, $\mathcal{C}(P_{tr})$ denotes the set of all community-induced subgraphs extracted from the training graph. Inspired by distributionally robust optimization (DRO) theory (Sagawa et al., 2019; Namkoong & Duchi, 2017), we focus on the worst-case community distribution $Q \in \mathcal{C}(P_{tr})$ that maximizes the loss discrepancy, which provides a principled way to capture heterogeneity: since different communities may correspond to distinct conditional distributions, optimizing for the hardest subgraph implicitly enforces stability across all latent communities. Thus, robustness to the worst-case subgraph translates into improved generalization under heterogeneous community shifts. This formu-

lation ensures that the learned predictor remains robust even when facing subgraphs with significant structural or feature deviations from the overall training distribution.

**Adversarial Perspective**  The key challenge in CHASM lies in the second term of our objective, i.e., identifying the hardest subgraph distribution $Q \in \mathcal{C}(P_{tr})$ and minimizing its discrepancy with the overall training distribution $P_{tr}$. To address this, we adopt an adversarial learning perspective. Specifically, we first pre-train the GNN for $k$ epochs to obtain initial node representations. Formally, given input features $X$ and adjacency $A$, the GNN encoder $f$ parameterized by $\theta$ produces $Z = f_\theta(X, A)$, where $Z \in \mathbb{R}^{N \times d}$ denotes the learned node embeddings. Based on these embeddings, CHASM formulates training as a two-player game: (1) a *Worst Community Extractor*, which adaptively mines the hardest subgraph $Q^*$ by maximizing the discrepancy $D_{\mathrm{KL}}(Q(Y|Z), P_{tr}(Y|Z))$, and (2) a *Community-Stability Learner*, which updates the GNN parameters $\theta$ to minimize the combined task loss and discrepancy penalty. To further strengthen the learner, we integrate an *Adaptive Subgraph Augmentation* module into this step: before computing the stability regularization, the mined hardest subgraph $Q^*$ is perturbed in both features and structure according to its distributional discrepancies with $P_{tr}$. This generates an augmented subgraph $\tilde{Q}^*$, ensuring that the learner is trained not only against the static worst-case distribution but also against more challenging yet realistic variants, thereby improving robustness.

## 3.2 WORST COMMUNITY EXTRACTOR

To characterize the most challenging regions of the training graph, we explicitly search for the worst-case community or subgraph whose predictive mechanism deviates most from the overall training distribution. Formally, given node representations $Z = f_\theta(X, A)$, the worst community distribution $Q^*$ is defined as:

$$Q^* = \arg \sup_{Q \in \mathcal{C}(P_{tr})} D_{\mathrm{KL}}\big(Q(Y|Z), P_{tr}(Y|Z)\big) = \arg \sup_{Q \in \mathcal{C}(P_{tr})} \mathbb{E}_{Z,Y \sim Q}\left[\log \frac{Q(Y|Z)}{P_{tr}(Y|Z)}\right], \quad (3)$$

where $\mathcal{C}(P_{tr})$ denotes the collection of all potential communities extracted from the training graph, $Q(Y|Z)$ is the conditional distribution restricted to subgraph $Q$, and $P_{tr}(Y|Z)$ is the conditional distribution under the overall training distribution. In practice, we parameterize both $Q(Y|Z)$ and $P_{tr}(Y|Z)$ using the prediction outputs of the current GNN, which enables tractable optimization. The *Worst Community Extractor* therefore acts as an adversary that adaptively mines the hardest subgraph $Q^*$ by maximizing its discrepancy with the training distribution. This mined subgraph serves as the focus of subsequent learning, ensuring that the model is consistently exposed to the most distribution-shifted regions of the graph.

For node classification with $K$ classes, given node representations $Z \in \mathbb{Z}$ and labels $Y \in [K]$, the conditional distribution $P(Y \mid Z)$ is discrete and can be modeled on the $K$-simplex:

$$P(Y \mid Z) := h_p(Z) \in \Delta_K, \quad (4)$$

where $h_p : \mathbb{Z} \to \Delta_K$ is a predictive mapping. In particular we denote the predictive distributions under the full training distribution and a candidate subgraph distribution by

$$P_{tr}(Y \mid Z) := h_{tr}(Z) \in \Delta_K, \qquad Q(Y \mid Z) := h_q(Z) \in \Delta_K, \quad (5)$$

where $h_{tr}$ and $h_q$ are the models fitted on $P_{tr}$ and $Q$, respectively. The worst community is defined as the distribution that maximizes the KL divergency between its predictive mechanism and the global one:

$$Q^* = \arg \sup_{Q \in \mathcal{C}(P_{tr})} \mathbb{E}_{Z,Y \sim Q}\left[\log \frac{h_q(Z)[Y]}{h_{tr}(Z)[Y]}\right], \quad (6)$$

where $h(Z)[Y]$ denotes the $Y$-th class probability of $h(Z) \in \Delta_K$.

To ensure the mined subgraph is structurally coherent rather than a scattered set of nodes, we incorporate a community-aware regularizer inspired by the Bernoulli–Poisson model. Let the mask generator be $m_\phi(Z)$ and write $F := m_\phi(Z)$ for the raw node scores. The negative log-likelihood of observing adjacency $A$ under the Bernoulli–Poisson interaction is

$$-\log p(A \mid F) = -\sum_{(u,v) \in E} \log\big(1 - \exp(-F_u^\top F_v)\big) + \sum_{(u,v) \notin E} F_u^\top F_v, \quad (7)$$

and, to avoid domination by non-edges in sparse graphs, we use a balanced empirical form

$$\mathcal{L}_{\text{com}} = -\mathbb{E}_{(u,v)\sim P_E}\left[\log\left(1 - \exp(-F_u^\top F_v)\right)\right] + \mathbb{E}_{(u,v)\sim P_N}\left[F_u^\top F_v\right], \tag{8}$$

where $P_E$ and $P_N$ denote uniform sampling over observed edges and sampled non-edges, respectively. This regularizer encourages high mask affinity for nodes that are likely to belong to the same community and penalizes spurious affinity across disconnected nodes.

To make the optimization in Eq. (6) practical, we adopt a weighted empirical formulation. Given the training nodes set $\{(x_i, y_i)\}_{i=1}^n$ drawn from $P_{tr}$, the training distribution can be represented by a uniform weight vector $\mathbf{w}_{tr} = [\frac{1}{n}, \ldots, \frac{1}{n}]^\top$. A candidate subgraph distribution is parameterized by a weight vector $\mathbf{w} = [w_1, \ldots, w_n]^\top \in \Delta_n$, where each $w_i$ represents the membership strength of node $i$ in the mined subgraph. The objective then becomes

$$\mathbf{w}^* = \arg\max_{\mathbf{w}\in\Delta_n}\ \sum_{i=1}^n w_i \cdot \log\frac{h_q(f_\theta(x_i))[y_i]}{h_{tr}(f_\theta(x_i))[y_i]}\ -\ \lambda_{\text{com}}\cdot\mathcal{L}_{\text{com}}, \tag{9}$$

where $\mathcal{L}_{\text{com}}$ is the community-aware loss in Eq. (8), ensuring structural coherence of the extracted subgraph, while $h_q$ is the classifier trained on the reweighted subgraph defined by $\mathbf{w}$.

Concretely, $h_{tr}$ and $h_q$ are obtained as

$$h_{tr} = \arg\min_h\ \sum_{i=1}^n \ell(h(f_\theta(x_i)), y_i),\ h_q = \arg\min_h\ \sum_{i=1}^n w_i\cdot\ell(h(f_\theta(x_i)), y_i). \tag{10}$$

Thus, the mining process is realized as a bi-level optimization: the inner problem updates $h_q$ given $\mathbf{w}$, while the outer problem adjusts $\mathbf{w}$ to maximize the discrepancy in (9) while simultaneously respecting community coherence. This results in an adversarially mined worst-case community $Q^*$ that is both predictive-hard and structurally plausible.

### 3.3 ADAPTIVE SUBGRAPH AUGMENTATION

To further improve the robustness of the learner against community-level shifts, we introduce an adaptive subgraph augmentation strategy. The key idea is to perturb the features and edges of the mined hardest subgraph, thereby exposing the model to more diverse patterns while maintaining task relevance. Unlike fixed perturbations, our augmentation intensity is adaptively controlled according to the discrepancies between the hardest subgraph and the overall training graph in both feature and structural spaces. Concretely, feature masking randomly removes a portion of node features, while edge dropout stochastically drops edges to simulate structural variability. To achieve more targeted augmentation, the perturbation probabilities are adaptively determined by the discrepancies between the hardest subgraph and the training graph: the feature distribution discrepancy is measured by the Euclidean distance between their means and standard deviations, while the structural discrepancy is measured by the difference in average node degrees. Based on these two metrics, we adaptively adjust the feature masking probability $p_{\text{feat}}$ and edge dropout probability $p_{\text{edge}}$, so that stronger perturbations are applied when larger discrepancies are detected. Formally, the augmentation module can be summarized as a transformation function:

$$(\tilde{X}, \tilde{A}) = g_{p_{\text{feat}}, p_{\text{edge}}}(X, A; Q^*), \tag{11}$$

where $Q^*$ is the mined hardest subgraph distribution and $g(\cdot)$ adaptively adjusts the perturbation strength. This design allows the learner to dynamically adapt augmentation intensity, enhancing generalization under both feature and structural distribution shifts.

### 3.4 COMMUNITY-STABILITY LEARNER

Given the hardest subgraph distribution $Q^*$ identified by the worst community extractor, the learner aims to minimize the classification loss while ensuring stability against the discrepancy between $Q^*$ and the overall training distribution $P_{tr}$. Formally, the objective of the learner is formulated as a Lagrangian penalty problem:

$$\mathcal{L}(\theta, \psi) = \mathbb{E}_{P_{tr}}\left[\ell\left(h_\psi(f_\theta(\tilde{X}, \tilde{A})), Y\right)\right] + \lambda\cdot D_{\text{KL}}\left(Q^*(Y \mid f_\theta(\tilde{X}, \tilde{A})) \parallel P_{tr}(Y \mid f_\theta(\tilde{X}, \tilde{A}))\right), \tag{12}$$

where $f_\theta(\cdot)$ is the GNN encoder parameterized by $\theta$, and $h_\psi(\cdot)$ is the classifier. The input $(\tilde{X}, \tilde{A})$ denotes the feature matrix and adjacency matrix after adaptive subgraph augmentation.

To make the KL-divergence term tractable, we adopt a first-order Taylor expansion approximation (Koyama & Yamaguchi, 2020), yielding:

$$D_{\mathrm{KL}}\big(Q^*(Y \mid f_\theta(\tilde{X}, \tilde{A})) \,\|\, P_{tr}(Y \mid f_\theta(\tilde{X}, \tilde{A}))\big) \approx \alpha\, \nabla_{\theta,\psi}\big(\mathcal{R}_{P_{tr}}(\theta, \psi) - \mathcal{R}_{\tilde{Q}^*}(\theta, \psi)\big)^\top \nabla_{\theta,\psi}\mathcal{R}_{\tilde{Q}^*}(\theta, \psi), \tag{13}$$

where $\alpha$ is the step size, $\mathcal{R}_{P_{tr}}$ and $\mathcal{R}_{\tilde{Q}^*}$ denote the average classification losses under $P_{tr}$ and the augmented hardest community $\tilde{Q}^*$, respectively. Finally, the overall optimization objective of the learner becomes:

$$\mathcal{L}(\theta, \psi) = \mathbb{E}_{P_{tr}}\Big[\ell\Big(h_\psi(f_\theta(\tilde{X}, \tilde{A})), Y\Big)\Big] + \lambda\, \nabla_{\theta,\psi}\big(\mathcal{L}_{P_{tr}}(\theta, \psi) - \mathcal{L}_{\tilde{Q}^*}(\theta, \psi)\big)^\top \nabla_{\theta,\psi}\mathcal{L}_{\tilde{Q}^*}(\theta, \psi), \tag{14}$$

which can be efficiently optimized via stochastic gradient descent. This design enables the learner to stabilize predictions across latent communities while adapting to feature and structural shifts through augmentation.

CHASM achieves robustness against community-level distribution shifts by jointly mining the hardest subgraphs, enforcing community-aware regularization, and augmenting them with adaptive perturbations. The adversarial interplay between the worst-community extractor and the community-stability learner ensures that the model not only fits the overall training distribution, but also remains stable under subgraphs exhibiting significant heterogeneity. The complete algorithm flow of CHASM is in Appendix 1

## 4 Theoretical Analysis

To theoretically justify the proposed CHASM framework, we conduct an analysis from the perspective of OOD generalization under latent community heterogeneity. Specifically, we study how the stability of predictive mechanisms across communities influences the generalization ability of the learned model, and establish a bound on the generalization error when shifting from training to unseen test communities.

**Assumption 4.1** (Community Expansion Assumption). *Let $f_\theta(X, A)$ be a graph representation and $h$ a predictor. We say the training-to-testing shift satisfies the community expansion assumption if there exists a monotone function $s : [0, \infty) \to [0, \infty)$ with $s(x) \geq x$ and $s(0) = 0$ such that for the learned representation $z = f_\theta(x, a)$,*

$$\mathbb{E}_{z \sim P_{te}(z)}\big[\mathrm{KL}(P_{te}(Y \mid z) \,\|\, P_{tr}(Y \mid z))\big] \leq s\left(\sup_{Q \in \mathcal{C}(P_{tr})} \mathrm{KL}(Q(Y \mid z) \,\|\, P_{tr}(Y \mid z))\right), \tag{15}$$

*where $\mathcal{C}(P_{tr})$ denotes the collection of community-induced sub-distributions of the training graph. The function $s(\cdot)$ captures how much the expected test conditional KL may be amplified relative to the worst-case KL observed among training communities.*

Building on this assumption, we now establish a formal generalization bound that connects community-level discrepancies in the training graph with the out-of-distribution (OOD) error on unseen test nodes.

**Theorem 4.2** (OOD generalization bound under community heterogeneity). *Assume the Community Expansion Assumption (15) holds for representation $f_\theta$ and predictor $h$. Let the loss $\ell(\cdot, \cdot)$ be bounded in $[0, M]$ for some $M > 0$. Then the conditional generalization error gap is bounded as*

$$\mathbb{E}_{z \sim P_{te}(z)}\Big[\big|\mathbb{E}_{P_{te}}[\ell(h(z), Y) \mid z] - \mathbb{E}_{P_{tr}}[\ell(h(z), Y) \mid z]\big|\Big]$$
$$\leq \mathcal{O}\Big(\sqrt{1 - \exp\big(-s\big(\sup_{Q \in \mathcal{C}(P_{tr})} \mathrm{KL}(Q(Y \mid z) \| P_{tr}(Y \mid z))\big)\big)}\Big). \tag{16}$$

**Remark.** Theorem 4.2 shows that the conditional generalization error on unseen communities is controlled by the square root of the community-expanded worst-case KL divergence measured among sufficiently large training subgraphs. Intuitively, if the predictive behavior remains consistent across training communities, then the bound guarantees smaller OOD generalization error. Conversely, large discrepancies across training communities enlarge the bound and indicate poor stability. This formal result provides a theoretical justification for CHASM: by explicitly mining hardest subgraphs, applying community-aware regularization, and performing adaptive augmentation, CHASM effectively reduces the worst-case divergence in training, thereby tightening the OOD generalization bound. The detailed proof is deferred to the appendix E.

Table 1: The performance on OOD benchmark datasets under **Covariate Shift**.

| Dataset | | CBAS | WebKB | Twitch | Cora | | Arxiv | | Require domain |
|---|---|---|---|---|---|---|---|---|---|
| Domain | | color | university | language | word | degree | time | degree | information |
| Base | ERM | 78.29±3.10 | 19.37±6.01 | 50.04±2.53 | 64.72±0.54 | 55.33±0.90 | 71.48±0.30 | 57.59±0.36 | No |
| Invariant | IRM | 78.00±3.73 | 21.59±9.30 | 50.97±3.62 | 64.70±0.45 | 55.34±0.88 | 71.36±0.13 | 57.54±0.15 | Yes |
| | VREx | 78.57±2.02 | 36.83±1.99 | 51.26±4.82 | 65.02±0.45 | 55.44±0.78 | 71.58±0.28 | 57.60±0.27 | Yes |
| DG | Coral | 78.00±3.59 | 30.16±5.67 | 53.46±0.41 | 64.85±0.33 | 55.21±0.90 | 71.48±0.40 | 57.23±0.16 | Yes |
| | DANN | 77.71±3.13 | 33.49±9.61 | 50.47±3.14 | 64.72±0.39 | 55.32±0.92 | 71.68±0.19 | 57.25±0.23 | No |
| DRO | KL-DRO | 77.71±1.63 | 32.06±16.96 | 53.76±4.19 | 64.85±0.51 | 55.14±0.87 | 71.58±0.24 | 57.59±0.30 | No |
| | GroupDRO | 77.14±2.67 | 25.24±9.57 | 50.48±2.04 | 64.95±0.59 | 55.05±0.83 | 71.46±0.33 | 57.25±0.23 | Yes |
| Graph OOD | SRGNN | 76.86±1.56 | 22.86±10.79 | 49.71±1.81 | 64.63±0.34 | 55.22±1.05 | 70.78±0.18 | 57.53±0.24 | Yes |
| | EERM | 75.14±3.29 | 33.97±11.42 | OOM | 64.88±0.38 | 55.30±0.89 | OOM | OOM | No |
| | Mixup | 70.43±5.28 | 17.78±2.11 | 56.58±0.77 | 65.05±0.51 | 57.53±0.78 | 71.27±0.22 | 57.42±0.24 | No |
| | FLOOD | 83.14±4.21 | 33.97±3.09 | 55.14±1.78 | 64.91±0.45 | 54.78±0.82 | 71.75±0.37 | 58.90±0.22 | No |
| | CIT | 80.86±2.96 | 28.89±9.09 | OOM | 64.34±0.72 | 54.74±0.82 | OOM | OOM | No |
| | CaNet | 76.71±2.12 | 31.96±5.74 | 50.72±0.50 | 63.12±0.43 | 55.95±0.43 | OOM | OOM | No |
| | CIA | 78.29±2.12 | 23.33±8.90 | 49.78±4.31 | 65.29±0.38 | 56.31±0.36 | 71.17±0.34 | 59.94±0.06 | No |
| **Ours** | **CHASM** | **87.14±1.08** | **38.89±5.08** | **60.07±1.89** | **66.20±0.50** | **58.62±0.40** | **71.88±0.16** | **60.50±1.25** | **No** |

Table 2: The performance on OOD benchmark datasets under **Concept Shift**.

| Dataset | | CBAS | WebKB | Twitch | Cora | | Arxiv | | Require domain |
|---|---|---|---|---|---|---|---|---|---|
| Domain | | color | university | language | word | degree | time | degree | information |
| Base | ERM | 82.43±2.56 | 27.16±0.93 | 51.59±3.63 | 64.03±0.28 | 60.30±0.46 | 65.64±0.27 | 54.81±0.40 | No |
| Invariant | IRM | 82.00±1.88 | 26.06±0.73 | 49.78±3.27 | 63.93±0.35 | 60.26±0.49 | 65.54±0.34 | 56.72±0.30 | Yes |
| | VREx | 82.86±2.42 | 26.61±1.42 | 55.75±1.37 | 64.03±0.29 | 60.53±0.39 | 65.92±0.14 | 56.68±0.35 | Yes |
| DG | Coral | 81.57±1.59 | 28.07±2.64 | 51.80±3.32 | 64.04±0.31 | 60.30±0.45 | 65.79±0.50 | 55.14±0.24 | Yes |
| | DANN | 83.57±1.52 | 29.36±3.31 | 51.67±3.50 | 63.96±0.29 | 60.23±0.51 | 65.67±0.42 | 55.34±0.45 | No |
| DRO | KL-DRO | 81.14±1.48 | 29.54±1.37 | 51.87±3.16 | 64.03±0.32 | 60.52±0.22 | 65.51±0.16 | 54.70±0.35 | No |
| | GroupDRO | 82.71±1.78 | 29.17±1.76 | 52.24±4.05 | 64.10±0.38 | 60.43±0.40 | 65.93±0.24 | 56.24±0.44 | Yes |
| Graph OOD | SRGNN | 82.14±2.12 | 26.42±1.78 | 51.58±3.64 | 63.96±0.34 | 60.27±0.35 | 65.64±0.34 | 55.08±0.25 | Yes |
| | EERM | 65.71±0.90 | 29.91±0.50 | OOM | 63.42±0.35 | 60.21±0.55 | OOM | OOM | No |
| | Mixup | 64.64±1.02 | 31.65±1.49 | 47.83±1.12 | 64.57±0.35 | 63.44±0.36 | 64.83±0.51 | 60.50±1.25 | No |
| | FLOOD | 84.29±1.82 | 28.62±6.29 | 54.22±3.92 | 64.01±0.26 | 60.31±0.48 | 65.66±0.26 | 58.59±1.56 | No |
| | CIT | 83.71±0.60 | 28.99±2.11 | OOM | 63.77±0.36 | 60.05±0.59 | OOM | OOM | No |
| | CaNet | 82.79±0.81 | 29.36±2.75 | 48.25±0.69 | 63.91±0.66 | 60.16±0.52 | OOM | OOM | No |
| | CIA | 83.86±2.60 | 31.93±2.78 | 50.40±3.95 | 64.52±0.57 | 61.32±0.47 | 65.99±0.18 | 58.65±0.31 | No |
| Ours | **CHASM** | **86.57±1.06** | **32.84±2.86** | **60.57±1.45** | **66.87±0.26** | **64.31±0.14** | **67.37±0.07** | **61.23±0.85** | **No** |

## 5 EXPERIMENTS

### 5.1 DATASETS AND BASELINES

**Datasets** We conduct experiments on five representative node classification datasets that exhibit both covariate and concept shifts: **WebKB** (Pei et al., 2020), **CBAS** (Ying et al., 2019), **Twitch** (Rozemberczki & Sarkar, 2020), **Cora** (Bojchevski & Günnemann, 2017), and **Arxiv** (Hu et al., 2020). Following the standard protocol of the GOOD benchmark (Gui et al., 2022), we use their official data splits, which are commonly adopted in recent studies (Wang et al., 2024; Guo et al., 2024). Basic statistics and further details of the datasets are provided in the Appendix D.1.

**Baselines** We evaluate the performance of our proposed CHASM framework against a diverse set of representative baselines, all using **GCN** (Kipf & Welling, 2017) as the shared backbone. These baselines include the standard Empirical Risk Minimization (**ERM**) strategy, along with two widely-used invariant learning methods: **IRM** (Arjovsky et al., 2019) and **VREx** (Krueger et al., 2021). We also consider two sample reweighting techniques, **KL-DRO** (Hu & Hong, 2013) and **Group DRO** (Sagawa et al., 2019), as well as two classical domain generalization algorithms: **DANN** (Ganin et al., 2016) and **Deep Coral** (Sun & Saenko, 2016). Furthermore, six graph-specific domain generalization methods are included for comparison: **EERM** (Wu et al., 2021), **SRGNN** (Zhu et al., 2021), **FLOOD** (Liu et al., 2023), **CIT** (Xia et al., 2024), **CaNet** (Wu et al., 2024), and **CIA** (Wang et al., 2024). The specific implementation details and parameter settings are provided in the Appendix D.2.

### 5.2 PERFORMANCE COMPARISON UNDER COVARIATE SHIFT AND CONCEPT SHIFT

We first evaluate CHASM and baselines under two major types of distribution shifts: *covariate shift* and *concept shift*, with results summarized in Tables 1 and 2. Under covariate shift, classical base-

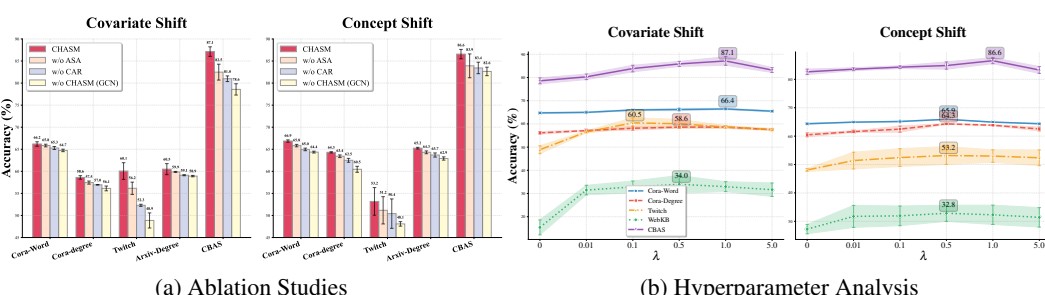

(a) Ablation Studies          (b) Hyperparameter Analysis

Figure 2: Ablation studies and hyperparameter sensitivity analysis of CHASM.

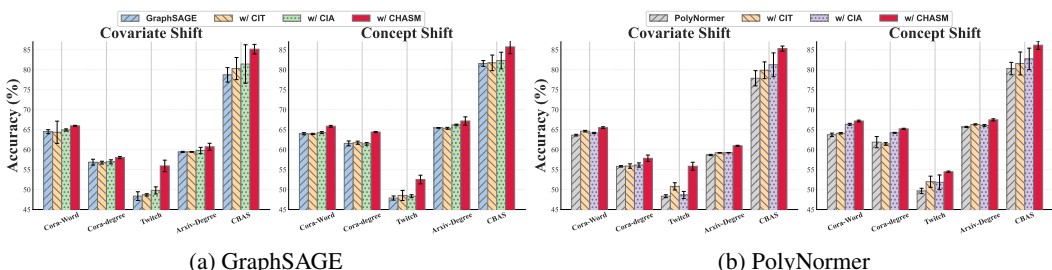

(a) GraphSAGE          (b) PolyNormer

Figure 3: Scalability analysis of CHASM under covariate and concept shifts on different backbones.

lines such as ERM and invariant learning methods fail to consistently improve robustness, while DG and DRO approaches bring partial gains but rely on explicit domain information. Existing graph OOD methods (e.g., EERM, CIA) achieve moderate improvements but suffer from instability and scalability issues. In contrast, CHASM achieves substantial gains across all datasets, often exceeding the strongest baseline by more than $+5\%$, showing that hard subgraph mining and adaptive augmentation effectively mitigate covariate perturbations. Under concept shift, where label mechanisms differ across environments, ERM, invariant learning, and DG methods again show limited robustness, and graph OOD methods yield inconsistent benefits. CHASM consistently outperforms all competitors, with notable improvements on WebKB and Cora, confirming its ability to enforce predictive stability across communities and adapt to unseen label mechanisms, thereby establishing new state-of-the-art performance under both types of distribution shifts.

**Take-away 1:** CHASM consistently improves performance under both covariate and concept shifts, showing that hard subgraph mining with community-aware regularization effectively enhances OOD robustness.

### 5.3 ABLATION STUDIES

To evaluate the contribution of each component in CHASM, we perform ablation studies by progressively removing modules: (i) w/o Adaptive Subgraph Augmentation (ASA), (ii) w/o ASA & Community-Aware Regularization (CAR), and (iii) replacing CHASM with a plain GCN backbone. Results in Figure 2a show that removing ASA already causes a clear drop under both covariate and concept shifts, while further removing CAR leads to larger degradation, especially on Arxiv and Twitch. The plain GCN suffers the most, confirming that neither augmentation nor structural alignment is preserved. ASA mainly improves robustness to covariate perturbations, CAR enforces community-level consistency, and their combination achieves the best OOD generalization.

**Take-away 2:** Overall, the ablation study demonstrates that each component of CHASM contributes positively to OOD robustness.

### 5.4 HYPERPARAMETER SENSITIVITY ANALYSIS

We further study the sensitivity of CHASM to the regularization weight $\lambda$ in the final optimization objective. Figure 2b shows the performance under both covariate and concept shift across five benchmark datasets when varying $\lambda$ from 0 to 5.0. We observe that a small positive value of $\lambda$

consistently improves performance compared to $\lambda = 0$, validating the effectiveness of the proposed community-aware regularization. As $\lambda$ increases, performance first improves and then stabilizes or slightly drops, with the best results typically achieved when $\lambda \in [0.1, 1.0]$. This indicates that CHASM is not overly sensitive to $\lambda$ and maintains robust performance across a relatively wide range, suggesting that the framework is stable and practical in real applications.

**Take-away 3:** CHASM achieves stable gains with $\lambda$ tuning, showing robustness and ease of use in practice.

### 5.5 SCALABILITY EXPERIMENTS

We evaluate CHASM's scalability by plugging it into two distinct GNN backbones (GraphSAGE (Hamilton et al., 2017) and the transformer-style PolyNormer (Deng et al., 2024)) and compare against strong OOD baselines (CIT (Xia et al., 2024), CIA (Wang et al., 2024)) under both covariate and concept shifts. As shown in Figure 3 (a)–(b), CHASM consistently yields the best or near-best performance across all datasets and shift types, while maintaining low run-to-run variance. Notably, the gains on the PolyNormer backbone indicate that our community-aware hard-subgraph mining and adaptive augmentation remain effective even for advanced, transformer-based graph architectures. These results demonstrate that CHASM is backbone-agnostic and scales well to different GNN designs, making it practically applicable on a wide range of graph models.

**Take-away 4:** CHASM is backbone-agnostic and scales effectively to diverse GNN architectures, ensuring robustness and adaptability in practice.

### 5.6 VISUALIZATION

To better understand what CHASM identifies as the hardest subgraphs, we visualize the WebKB dataset under both **covariate shift** and **concept shift**. Unlabeled nodes are shown in gray, training nodes in blue, and the mined hardest subgraph in red. As shown in Figure 4, the red nodes form compact clusters with clear community boundaries, which is consistent with the inherent community structure of WebKB. This indicates that CHASM does not simply capture ran-

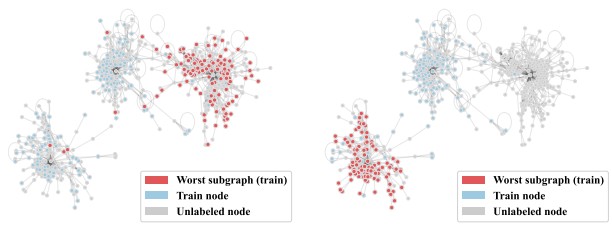

(a) Concept Shift.  (b) Covariate Shift.

Figure 4: Visualization of WebKB under concept and covariate shift. Gray: unlabeled nodes, Blue: training nodes, Red: hardest subgraph mined by CHASM.

dom noisy samples, but instead mines structurally coherent and distributionally challenging subgraphs. Comparing the two types of shifts, we observe that under covariate shift the mined subgraphs are more strongly associated with structural variations, while under concept shift they reflect regions with more label uncertainty, highlighting CHASM's ability to adaptively capture different sources of difficulty.

**Take-away 5:** CHASM identifies and visualizes structurally coherent hardest subgraphs under various distribution shifts, providing valuable insights into the sources of OOD challenges in graphs, which can guide the development of targeted robustness-enhancing strategies.

## 6 CONCLUSION

In this paper, we proposed CHASM, a community-aware framework for OOD generalization on graphs. By adversarially mining hardest subgraphs, enforcing structural coherence, and applying adaptive augmentation, CHASM explicitly targets community-level shifts. A stability-based learner with theoretical guarantees further enhances robustness. Extensive experiments under covariate and concept shifts, together with ablation and scalability studies, demonstrate the consistent superiority and interpretability of CHASM across diverse benchmarks.

## REPRODUCIBILITY STATEMENT

We are committed to ensuring the reproducibility of our work. All resources necessary to replicate our results are provided in the main text and appendix. Specifically, the full implementation of CHASM, including training scripts and hyperparameter settings, is available in the appendix. All theoretical motivations and assumptions are clearly presented in Section 4, with formal derivations provided in Appendix E. The benchmark datasets used in our experiments, along with detailed data splits, are described in Section 5.1 and Appendix D. To facilitate further verification, we also include additional experimental results in Appendix D. Together, these resources ensure that all reported results can be faithfully reproduced.

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

# A    RELATED WORKS

## A.1    OUT-OF-DISTRIBUTION GENERALIZATION

Out-of-distribution (OOD) generalization studies how to train models that remain reliable when the test distribution differs from the training distribution. Two dominant families of approaches in the literature are *invariant learning* and *distributionally robust optimization* (DRO).

Invariant learning aims to find features or predictors whose predictive relations to labels remain stable across multiple environments. Classical works such as Invariant Risk Minimization (IRM) and variance-regularized variants encourage consistency of predictors across training domains (Arjovsky et al., 2019; Krueger et al., 2021). Recent developments formalize weaker or more practical stability notions and design algorithms that explicitly penalize instability of prediction mechanisms among sub-populations; a representative example is Stable Risk Minimization (SRM) (Liu et al., 2024a), which quantifies and optimizes a sub-population stability penalty to improve robustness under mechanism shifts. DRO instead frames OOD robustness as a worst-case optimization over an ambiguity set of distributions. DRO methods provide a principled route to guard against adversarial or worst-case shifts and come with theoretical guarantees under carefully chosen ambiguity sets (e.g., KL- or Wasserstein-based constraints) (Duchi & Namkoong, 2018; Sagawa et al., 2019). However, naively applying DRO can suffer from over-pessimism (overweighting noisy or outlier samples) and practical inefficiencies. Recent work such as Geometry-Calibrated DRO (Liu et al., 2024b) proposes geometry-aware calibration to mitigate over-pessimistic behavior and better respect the underlying data manifold. While invariant learning and DRO have complementary strengths, several practical challenges remain when adapting them to graph-structured data. Graph data exhibit relational dependencies and heterogeneous community structure that complicate environment construction and ambiguity-set design; naive augmentations or domain partitions may break structural coherence and produce unrealistic environments.

## A.2    GRAPH OUT-OF-DISTRIBUTION GENERALIZATION

Recent progress in graph invariant learning has shown strong potential for addressing out-of-distribution (OOD) challenges in graph-level tasks (Wu et al., 2022; Li et al., 2022; Yang et al., 2022; Sui et al., 2023). These approaches generally aim to capture stable relationships between node features, structural patterns, and labels while filtering out spurious correlations caused by distribution shifts. For instance, GIL (Li et al., 2022) extracts invariant structural–label relationships across latent environments, while DIR (Wu et al., 2022) constructs diverse distributions via augmentation guided by causal rationales. MoleOOD (Yang et al., 2022) focuses on inferring latent environments in molecular graphs, and AIA (Sui et al., 2023) leverages adversarial augmentation to generate new training environments. Li et al. (Li et al., 2024) propose graph structure extrapolation to generate challenging yet realistic OOD graphs. While effective, these methods mainly focus on graph-level prediction tasks, and their reliance on handcrafted or augmented environments often limits their adaptability to more fine-grained node-level shifts.

Compared to graph-level tasks, node-level OOD generalization remains less explored but equally critical, as distribution shifts can manifest both in node features and structural neighborhoods. Early works such as EERM (Wu et al., 2021) train adversarial context explorers to construct virtual environments for invariant representation learning, while CIT (Xia et al., 2024) improves node diversity by generating samples across clusters. FLOOD (Liu et al., 2023) combines invariant and bootstrapped learning to improve robustness under heterogeneous shifts. TAR's topology-aware reweighting (Zheng et al., 2024) and LRW-OOD's learnable random walk (Sun et al., 2025) demonstrate the importance of topology-aware exploration, while DeCaf (Han et al., 2024) introduces causal modeling to decouple distribution factors at the node level. Despite this progress, existing methods often rely on either synthetic environments of limited reliability or strong invariance assumptions. Moreover, they rarely account for noisy or adversarial subgraphs that exacerbate distributional heterogeneity. Our proposed CHASM framework addresses these limitations by explicitly mining hardest subgraphs that reflect latent community structures, applying community-aware regularization, and adaptively augmenting discrepancies, thereby providing a more unified and flexible approach to node-level OOD generalization.

# B  OTHER PRELIMINARIES

## B.1  DISTRIBUTION SHIFT TYPE.

Following GOOD (Gui et al., 2022), we formalize the two types of distribution shift in node classification. For graph data, the joint distribution can be decomposed as $\mathbb{P}(Y, X, A) = \mathbb{P}(Y \mid X, A)\mathbb{P}(X, A)$, where $X$ and $A$ denote node features and graph structure, respectively. Accordingly, the main types of shifts are:

1. **Covariate Shift** ($\mathbb{P}_s(Y \mid X, A) = \mathbb{P}_t(Y \mid X, A)$, $\mathbb{P}_s(X, A) \neq \mathbb{P}_t(X, A)$): In this case, the conditional label mechanism is invariant across domains, while the joint distribution of node attributes and structural connectivity patterns changes. Such shifts reflect variations in the feature space or graph topology that alter the marginal distribution of the input, without affecting the relationship between input and label.

2. **Concept Shift** ($\mathbb{P}_s(Y \mid X, A) \neq \mathbb{P}_t(Y \mid X, A)$, $\mathbb{P}_s(X, A) = \mathbb{P}_t(X, A)$): Here, the marginal distribution of node features and structures remains stable across domains, but the conditional relationship between input and label differs. This corresponds to changes in the predictive mechanism itself, where the same node attributes or structural properties may imply different labels in different domains.

## B.2  GRAPH NEURAL NETWORK.

Graph Neural Networks (GNNs) have emerged as the standard for learning on graph-structured data, due to their ability to capture both node features and topological dependencies. A typical GNN layer can be abstracted into two stages: message aggregation (**AGGR**($\cdot$)) and representation update (**UPD**($\cdot$)). Formally, the representation of node $i$ at layer $l + 1$ is computed as:

$$z_i^{(l+1)} = \mathbf{UPD}\Big(z_i^{(l)}, \mathbf{AGGR}\big(\{z_j^{(l)} \mid j \in \mathcal{N}(i)\}\big)\Big), \tag{17}$$

where $\mathcal{N}(i)$ denotes the set of neighbors of node $i$. Different GNN variants mainly differ in the choice of **AGGR** and **UPD**, such as mean or attention-based aggregation and linear or nonlinear transformation for updates. By stacking multiple layers, GNNs enable nodes to integrate multi-hop neighborhood information and learn expressive latent representations.

In this work, we focus on the node classification task, where the goal is to predict the label of each node based on its features and structural context. While GNNs have shown remarkable success in this setting, their performance often deteriorates under distribution shifts between training and testing nodes. The central objective of our study is therefore to improve the out-of-distribution generalization of GNN models by explicitly addressing latent community heterogeneity.

# C  ALGORITHM

Algorithm 1 provides a detailed depiction of the training workflow.

# D  EXPERIMENTAL DETAILS

## D.1  DATASET

In this paper, we utilize four 4 classification datasets subjected to both concept shift and covariate shift from GOOD benchmark (Gui et al., 2022). They are respectively **Cora** (Bojchevski & Günnemann, 2017), **Arxiv** (Hu et al., 2020), **WebKB** (Pei et al., 2020), and **Twitch** (Rozemberczki & Sarkar, 2020). The detailed information about the dataset construction is as follows:

- **CBAS** is a synthetic dataset where the input graph is constructed by attaching 80 house-like motifs to a 300-node Barabási–Albert base graph. The task involves predicting the role of each node, including whether a node is the top, middle, or bottom of a house-like motif, or belongs to the base graph, resulting in a 4-class classification task. In CBAS, different node colors are used as features, so OOD algorithms must address both covariate shifts caused by node color differences and concept shifts driven by color-label correlations.

---

**Algorithm 1** CHASM: Community-Aware Hard Subgraph Mining

---

**Require:** Training Graph $G = (X, A, Y)$, hyperparameters $\lambda, \lambda_{\text{com}}$, epochs $T$
**Ensure:** Trained GNN model $h_\psi(f_\theta(\cdot))$
1: **Initialization:** Train $f_\theta$ on $(X, A)$ for $k$ warm-up epochs to obtain initial representations.
2: **for** $t = k+1, k+2, \ldots, T$ **do**
3:    **Worst Community Extracting**
4:    Compute node weights via mask model $m_\phi(X)$;
5:    Extract hardest subgraph distribution $Q^*$ maximizing predictive discrepancy:
6:       $Q^* = \arg\max_Q \; \mathbb{E}_{(x,y)\sim Q}\left[\log \frac{f_q(x)[y]}{f_{tr}(x)[y]}\right]$;
7:    Apply community loss $\mathcal{L}_{com}$ to enforce structural coherence.
8:    **Adaptive Subgraph Augmentation:**
9:    Adjust perturbation strengths $p_{\text{feat}}, p_{\text{edge}}$ and generate augmented subgraph $(\tilde{X}, \tilde{A}) = g_{p_{\text{feat}}, p_{\text{edge}}}(X, A; Q^*)$.
10:    **Community-Stability Learning (Learner)**
11:    Update parameters by minimizing:
12:       $\mathcal{L}(\theta, \psi) = \mathbb{E}_{P_{tr}}[\ell(h_\psi(f_\theta(\tilde{X}, \tilde{A})), Y)] + \lambda \nabla_{\theta,\psi}(\mathcal{L}_{P_{tr}} - \mathcal{L}_{\tilde{Q}^*})^\top \nabla_{\theta,\psi}\mathcal{L}_{\tilde{Q}^*}.$
13: **end for**
14: **return** $h_\psi(f_\theta(\cdot))$

---

Table 3: Numbers of nodes in training, ID validation, ID test, OOD validation, and OOD test sets for the four datasets.

| Dataset | Shift | Train | ID Validation | ID test | OOD Validation | OOD Test |
|---|---|---|---|---|---|---|
| Cora-Word | concept | 7,273 | 1,558 | 1,558 | 3,807 | 5,597 |
| | covariate | 9,378 | 1,979 | 1,979 | 3,003 | 3,454 |
| Cora-Degree | concept | 7,281 | 1,560 | 1,560 | 3,706 | 5,686 |
| | covariate | 8,213 | 1,979 | 1,979 | 3,841 | 3,781 |
| Arxiv-Time | concept | 62,803 | 13,303 | 13,303 | 32,560 | 48,094 |
| | covariate | 57,073 | 16,934 | 16,934 | 29,799 | 48,603 |
| Arxiv-Degree | concept | 58,619 | 12,561 | 12,561 | 34,222 | 51,380 |
| | covariate | 68,607 | 16,934 | 16,934 | 46,264 | 20,604 |
| Twitch | concept | 13,605 | 2,914 | 2,914 | 6,762 | 7,925 |
| | covariate | 14,448 | 3,412 | 3,412 | 6,551 | 6,297 |
| WebKB | concept | 282 | 60 | 60 | 106 | 109 |
| | covariate | 244 | 61 | 61 | 125 | 126 |
| CBAS | concept | 140 | 140 | 140 | 140 | 140 |
| | covariate | 420 | 70 | 70 | 70 | 70 |

- **GOOD-Cora** is a citation network dataset where nodes represent scientific publications and edges represent citation links. The task is to classify publications into 70 categories. Splits are based on word diversity and node degree, ensuring classifications are not influenced by irrelevant factors.

- **GOOD-Arxiv** is a citation dataset representing a directed graph of computer science arXiv papers and their citations. The task is to classify papers into 40 subject areas. Splits are based on publication year and node degree.

- **GOOD-Twitch** is a gamer network dataset where nodes represent gamers, with games as features, and edges as friendships. The task is a binary classification to predict if a user streams mature content. Splits are based on user language.

- **GOOD-WebKB** is a university webpage network dataset where nodes represent webpages, with words on the webpage as features, and edges as hyperlinks. The task is to classify webpages into five categories. Splits are based on the domain university, focusing on word content and link connections.

Regarding the dataset splitting, we used the splits provided by the GOOD benchmark. The specific split details are shown in Table 3. For more detailed information on dataset splitting, please refer to (Gui et al., 2022).

## D.2 IMPLEMENTATION DETAILS

To ensure a fair comparison, we integrated our proposed CHASM framework into the GOOD Benchmark, eliminating potential inconsistencies caused by different training pipelines. We followed the standard dataset splits provided by the benchmark. All experiments were conducted with 10 random seeds, and each experiment was repeated 5 times; the averages and standard deviations were reported. Identical seeds were used for all baselines and our method to avoid randomness caused by initialization.

For all OOD algorithms, we adopted a shared backbone consisting of a 3-layer GCN encoder and a 1-layer MLP classifier. The hidden dimension of the GCN was fixed at 256, with ReLU activation and a dropout rate of 0.5. Training was performed with the Adam optimizer, an initial learning rate of 0.001, and weight decay of $5 \times 10^{-5}$. The batch size was set to 256, and the number of epochs was fixed at 200 for small datasets (e.g., Cora, WebKB) and 400 for larger datasets (e.g., Arxiv, Twitch). Early stopping was applied with a patience of 50 epochs based on validation accuracy.

For CHASM-specific hyperparameters, we tuned the stability regularization weight $\lambda \in \{0.01, 0.05, 0.1, 0.5, 1.0\}$ and set it to 0.5 unless otherwise specified. The mask model $m_\phi(\cdot)$ for subgraph mining was implemented as a 2-layer MLP with hidden size 128 and sigmoid gating. For adaptive subgraph augmentation, the feature masking probability $p_{\text{feat}}$ and edge dropout probability $p_{\text{edge}}$ were dynamically adjusted within the range $[0.1, 0.5]$ according to discrepancies in feature and structural statistics between the mined hardest subgraph and the overall training graph. The coefficient for community-aware regularization was set to 0.1.

All baselines and CHASM were tuned via grid search following the GOOD Benchmark protocol, and their best validation performance was reported. Note that certain memory-intensive algorithms could not scale to large datasets like Arxiv due to GPU memory limits.

- **Hardware:** NVIDIA RTX 3090 GPU with 24GB memory.
- **Software:** Python 3.8, PyTorch 1.10.1, PyG 2.0.3, CUDA 11.3.
- **Optimizer:** Adam, learning rate = 0.001, weight decay = $5 \times 10^{-5}$.
- **Backbone:** 3-layer GCN encoder + 1-layer MLP classifier, hidden size = 256, dropout = 0.5.
- **Training:** batch size = 256; epochs = 200 (small datasets) / 400 (large datasets); early stopping patience = 150.

## D.3 PERFORMANCE COMPARISON UNDER IN-DOMAIN SCENARIO

To further validate the effectiveness of CHASM, we conduct experiments on **in-domain test samples** under both covariate and concept shift settings, with results reported in Tables 4 and 5. Overall, most methods, including ERM and invariant learning baselines, achieve relatively strong performance in the in-domain setup, as expected. Interestingly, Mixup achieves competitive or even better results on specific datasets such as WebKB and Cora, which can be attributed to its strong regularization effect under limited distributional variation. However, such improvements are not consistent and often come with instability across different datasets.

In contrast, **CHASM consistently delivers the best or near-best performance across almost all datasets and shift types**, demonstrating its robustness even in the in-domain regime. This indicates that the hard subgraph mining and adaptive augmentation strategies of CHASM not only help mitigate distributional discrepancies under OOD settings but also enhance the stability and generalization of the model within the training domain. These results confirm that CHASM provides a principled and effective framework that benefits both in-domain and OOD generalization.

## D.4 ADDITIONAL SCALABILITY EXPERIMENTS

To further examine the scalability of CHASM, we extend our analysis to additional backbone architectures, including **GAT** (Veličković et al., 2018), **SGC** (Wu et al., 2019), **GATv2** (Brody et al., 2021), and **GIN** (Xu et al., 2018). The results under both covariate and concept shifts are reported in Figure 5. Across all these backbones, CHASM consistently outperforms the corresponding baselines, showing notable gains particularly on challenging datasets such as CBAS and Twitch. These

Table 4: In-domain performance under **Covariate Shift**. We report the average test accuracy (except for Twitch, where ROC-AUC is used) and standard deviations over 5 runs. The best results are shown in **bold**. OOM denotes out of memory.

| Dataset | | CBAS | WebKB | Twitch | Cora | | Arxiv | | Require domain information |
| --- | --- | --- | --- | --- | --- | --- | --- | --- | --- |
| Domain | | color | university | language | word | degree | time | degree | |
| Base | ERM | 93.71±1.83 | 41.31±5.43 | 72.44±0.20 | 70.39±0.33 | 72.50±0.48 | 72.60±0.14 | 77.28±0.25 | No |
| Invariant | IRM | 92.14±2.87 | 44.10±5.65 | 72.10±0.79 | 70.38±0.38 | 72.78±0.44 | 72.49±0.21 | 77.23±0.29 | Yes |
| | VREx | 92.00±2.87 | 40.98±12.42 | 72.82±6.41 | 69.89±0.29 | 72.19±0.48 | 70.29±0.33 | 77.12±0.25 | Yes |
| DG | Coral | 92.71±2.74 | 43.11±4.58 | 72.38±0.15 | 70.50±0.37 | 72.53±0.49 | 72.50±0.14 | 77.20±0.25 | Yes |
| | DANN | 91.00±2.64 | 42.30±5.17 | 72.11±0.59 | 70.45±0.27 | 72.52±0.47 | 72.60±0.16 | 77.22±0.25 | No |
| DRO | KL-DRO | 92.24±1.11 | 42.95±2.14 | 72.30±2.44 | 69.75±0.50 | 72.68±0.19 | 72.56±0.80 | 76.19±0.67 | No |
| | GroupDRO | 92.57±2.00 | 44.92±4.10 | 72.07±0.52 | 70.54±0.34 | 72.33±0.52 | 72.51±0.10 | 76.98±0.24 | Yes |
| Graph OOD | SRGNN | 79.43±2.80 | 44.92±6.85 | 72.41±0.44 | 70.08±0.44 | 72.28±0.53 | 72.64±0.18 | 77.68±0.33 | Yes |
| | EERM | 64.00±3.93 | 34.43±7.48 | OOM | 69.19±0.40 | 72.45±0.55 | OOM | OOM | No |
| | Mixup | 73.24±5.21 | **55.41±3.50** | 72.39±0.43 | **72.04±0.55** | 74.03±0.42 | 72.36±0.14 | 77.39±0.17 | No |
| | FLOOD | 92.29±1.46 | 52.46±6.89 | 70.18±3.79 | 70.36±0.70 | 73.43±0.42 | 72.11±0.98 | 77.69±0.96 | No |
| | CIT | 87.28±2.43 | 41.47±4.02 | OOM | 70.22±0.58 | 72.59±0.68 | OOM | OOM | No |
| | CaNet | 80.57±1.00 | 42.06±0.79 | 70.50±0.30 | 69.64±0.48 | 72.48±0.59 | OOM | OOM | No |
| | CIA | 90.00±1.43 | 43.93±1.37 | 71.93±0.90 | 70.29±0.49 | 73.35±0.40 | 72.90±0.36 | 77.59±0.11 | No |
| **Ours** | **CHASM** | **94.29±1.75** | 49.51±2.69 | **74.71±0.09** | 71.72±0.27 | **74.44±0.35** | **73.91±0.17** | **78.68±0.14** | **No** |

Table 5: In-domain performance under **Concept Shift**. We report the average test accuracy (except for Twitch, where ROC-AUC is used) and standard deviations over 5 runs. The best results are shown in **bold**. OOM denotes out of memory.

| Dataset | | CBAS | WebKB | Twitch | Cora | | Arxiv | | Require domain information |
| --- | --- | --- | --- | --- | --- | --- | --- | --- | --- |
| Domain | | color | university | language | word | degree | time | degree | |
| Base | ERM | 90.07±1.26 | 62.17±3.17 | 82.96±0.43 | 66.15±0.48 | 68.36±0.39 | 73.75±0.34 | 74.65±0.39 | No |
| Invariant | IRM | 89.64±1.16 | 61.33±2.67 | 82.91±0.48 | 66.20±0.27 | 68.47±0.32 | 74.01±0.24 | 74.73±0.28 | Yes |
| | VREx | 90.50±1.20 | 61.00±3.27 | 81.03±3.36 | 65.97±0.39 | 68.31±0.15 | 73.95±0.27 | 73.00±0.60 | Yes |
| DG | Coral | 89.22±0.67 | 61.17±3.42 | 82.93±0.43 | 66.10±0.34 | 68.47±0.24 | 74.02±0.28 | 74.73±0.25 | Yes |
| | DANN | 90.07±1.58 | 61.67±2.36 | 83.01±0.44 | 66.19±0.40 | 68.45±0.28 | 73.85±0.27 | 74.82±0.33 | No |
| DRO | KL-DRO | 89.29±0.51 | 62.67±1.49 | 83.23±0.11 | 66.14±0.16 | 67.57±0.54 | 74.07±0.24 | 75.13±0.20 | No |
| | GroupDRO | 90.57±1.19 | 60.33±2.67 | 83.05±0.39 | 66.17±0.49 | 68.36±0.39 | 73.60±0.79 | 74.91±0.26 | Yes |
| Graph OOD | SRGNN | 89.14±1.27 | 61.50±1.89 | 83.01±0.36 | 66.11±0.12 | 68.33±0.41 | 73.45±0.44 | 73.44±0.08 | Yes |
| | EERM | 78.50±1.85 | 61.50±7.33 | OOM | 65.57±0.49 | 66.33±0.75 | OOM | OOM | No |
| | Mixup | 93.64±1.33 | 70.17±2.36 | 80.84±1.96 | **69.65±0.72** | 70.23±0.50 | 74.68±0.28 | 72.44±0.89 | No |
| | FLOOD | 90.07±1.68 | 62.50±2.71 | 83.22±0.94 | 66.15±0.32 | 68.92±0.72 | 72.11±0.98 | 70.69±0.96 | No |
| | CIT | 91.00±1.08 | 62.00±2.49 | OOM | 66.49±0.39 | 68.09±0.46 | OOM | OOM | No |
| | CaNet | 89.50±0.91 | 55.84±1.83 | 82.24±0.52 | 66.35±0.53 | 68.74±0.52 | OOM | OOM | No |
| | CIA | 91.86±1.30 | 65.11±4.80 | 83.93±0.17 | 66.20±0.14 | 67.59±0.37 | 73.86±0.48 | 74.74±0.31 | No |
| Ours | **CHASM** | **94.14±2.78** | **70.67±0.91** | **85.33±0.16** | 68.41±0.62 | 68.97±0.3 | **76.99±0.11** | **77.48±0.32** | **No** |

results confirm that the effectiveness of CHASM is not limited to specific GNN architectures, but generalizes well across message-passing, attention-based, and simplified propagation models. This demonstrates the strong adaptability of CHASM to diverse GNN designs, further validating its practical applicability in real-world scenarios.

### D.5 Loss Curve Visualization

To further investigate the training dynamics of CHASM, we visualize the loss curves across several datasets under both covariate and concept shifts, as shown in Figure 6. Although CHASM adopts an adversarial training scheme, which is often criticized for instability and convergence issues, the curves demonstrate that our framework is ultimately stable. Specifically, during the early stage of training, when the model begins adversarially mining the hardest subgraphs, noticeable oscillations appear in the validation and test losses. This is expected, as the adversary actively searches for the most challenging community distributions. However, after sufficient iterations, the losses consistently converge to a stable level across all datasets and settings. These results verify that despite the initial fluctuations, CHASM achieves reliable convergence, further supporting its practicality for robust graph learning under distribution shifts.

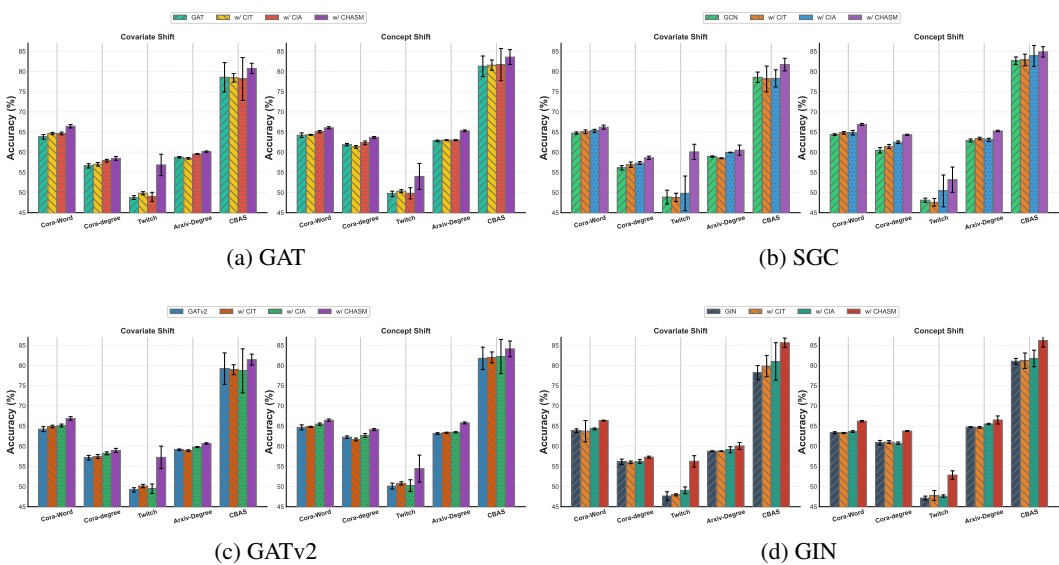

Figure 5: Scalability analysis of CHASM under covariate and concept shifts on more different backbones.

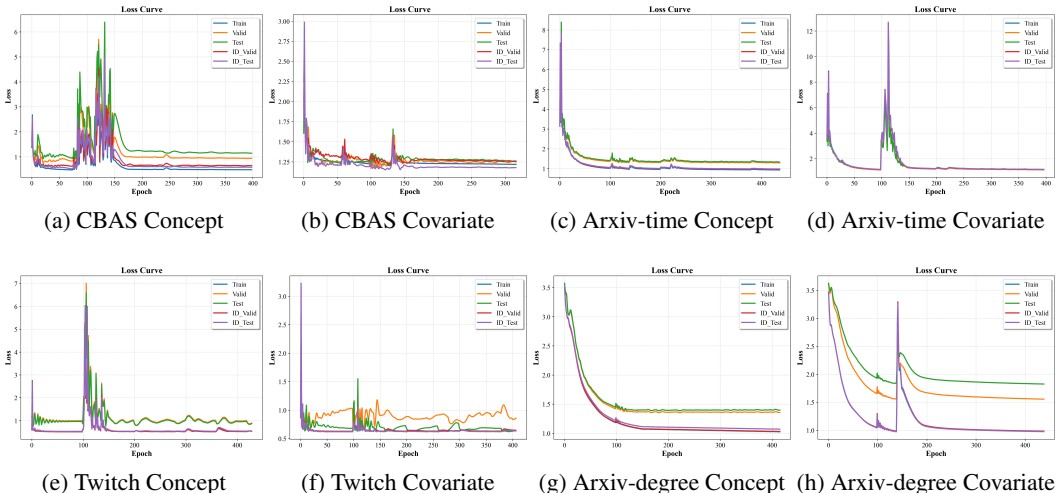

Figure 6: Loss curve visualization of CHASM across multiple datasets under covariate and concept shifts. Although adversarial training introduces oscillations in the early stages due to hardest subgraph mining, the curves eventually converge stably, demonstrating the reliable convergence of CHASM.

## E  PROOF OF THEOREM

*Proof.* Fix any representation value $z = f_\theta(x, a)$. Define $L_z(y) := \ell(h(z), y)$ with $\ell(\cdot, \cdot) \in [0, M]$, and $g_z(y) := L_z(y)/M \in [0, 1]$.

$$\Delta_\ell(z) = \big| \mathbb{E}_{P_{te}}[L_z(Y) \mid z] - \mathbb{E}_{P_{tr}}[L_z(Y) \mid z] \big| = M \big| \mathbb{E}_{P_{te}}[g_z(Y) \mid z] - \mathbb{E}_{P_{tr}}[g_z(Y) \mid z] \big|. \quad (18)$$

Using $g_z(y) = \int_0^1 \mathbf{1}\{g_z(y) > t\} \, dt$ and Fubini's theorem,

$$\mathbb{E}_P[g_z(Y) \mid z] = \int_0^1 P(g_z(Y) > t \mid z)\, dt, \quad P \in \{P_{te}, P_{tr}\}, \tag{19}$$

$$
\begin{aligned}
\left| \mathbb{E}_{P_{te}}[g_z(Y) \mid z] - \mathbb{E}_{P_{tr}}[g_z(Y) \mid z] \right| &\leq \int_0^1 \left| P_{te}(g_z(Y) > t \mid z) - P_{tr}(g_z(Y) > t \mid z) \right| dt \\
&\leq \mathrm{TV}(P_{te}(Y \mid z), P_{tr}(Y \mid z)),
\end{aligned} \tag{20}
$$

$$\Delta_\ell(z) \leq M \cdot \mathrm{TV}(P_{te}(Y \mid z), P_{tr}(Y \mid z)). \tag{21}$$

By Pinsker's inequality, $\mathrm{TV}(P, Q) \leq \sqrt{\frac{1}{2}\mathrm{KL}(P\|Q)}$,

$$\Delta_\ell(z) \leq M\sqrt{\frac{1}{2}\mathrm{KL}(P_{te}(Y \mid z)\|P_{tr}(Y \mid z))}. \tag{22}$$

Taking expectation over $z \sim P_{te}(z)$, assuming covariate shift is controlled and $P_{te}(z)$ extends $P_{tr}(z)$ with bounded discrepancy,

$$\mathbb{E}_{z\sim P_{te}}[\Delta_\ell(z)] \leq M\mathbb{E}_{z\sim P_{te}}\left[ \sqrt{\frac{1}{2}\mathrm{KL}(P_{te}(Y \mid z)\|P_{tr}(Y \mid z))} \right]. \tag{23}$$

Since $\sqrt{\cdot}$ is concave, by Jensen's inequality,

$$\mathbb{E}_{z\sim P_{te}}\left[ \sqrt{\mathrm{KL}(P_{te}(Y \mid z)\|P_{tr}(Y \mid z))} \right] \leq \sqrt{\mathbb{E}_{z\sim P_{te}}\left[ \mathrm{KL}(P_{te}(Y \mid z)\|P_{tr}(Y \mid z)) \right]}, \tag{24}$$

$$\mathbb{E}_{z\sim P_{te}}[\Delta_\ell(z)] \leq M\sqrt{\frac{1}{2}\mathbb{E}_{z\sim P_{te}}\left[ \mathrm{KL}(P_{te}(Y \mid z)\|P_{tr}(Y \mid z)) \right]}. \tag{25}$$

By the Community Expansion Assumption (Eq. (15)), with the supremum KL bounded by the $\alpha_0$-proportion constraint in $\mathcal{C}(P_{tr})$,

$$\mathbb{E}_{z\sim P_{te}}\left[ \mathrm{KL}(P_{te}(Y \mid z)\|P_{tr}(Y \mid z)) \right] \leq s\left( \sup_{Q\in\mathcal{C}(P_{tr})} \mathrm{KL}(Q(Y \mid z)\|P_{tr}(Y \mid z)) \right), \tag{26}$$

$$\mathbb{E}_{z\sim P_{te}}[\Delta_\ell(z)] \leq M\sqrt{\frac{1}{2}s\left( \sup_{Q\in\mathcal{C}(P_{tr})} \mathrm{KL}(Q(Y \mid z)\|P_{tr}(Y \mid z)) \right)}. \tag{27}$$

Let $S = s\left( \sup_{Q\in\mathcal{C}(P_{tr})} \mathrm{KL}(Q(Y \mid z)\|P_{tr}(Y \mid z)) \right) \geq 0$. For $s \in [0, S_0]$ (where $S_0$ is bounded by data complexity),

$$0 \leq 1 - e^{-s} \leq s, \quad \sqrt{1 - e^{-s}} \leq \sqrt{s}, \tag{28}$$

and since $\phi(s) = (1 - e^{-s})/s \geq \phi(S_0) > 0$ for $s \in (0, S_0]$,

$$1 - e^{-s} \geq \phi(S_0)s, \quad \sqrt{s} \leq \frac{1}{\sqrt{\phi(S_0)}}\sqrt{1 - e^{-s}}, \tag{29}$$

$$M\sqrt{\frac{1}{2}S} = \mathcal{O}\left( \sqrt{1 - \exp\left( -s\left( \sup_{Q\in\mathcal{C}(P_{tr})} \mathrm{KL}(Q(Y \mid z)\|P_{tr}(Y \mid z)) \right) \right)} \right). \tag{30}$$

Thus, the conditional generalization error gap is bounded as

$$\mathbb{E}_{z \sim P_{te}(z)}\left[\left|\mathbb{E}_{P_{te}}[\ell(h(z), Y) \mid z] - \mathbb{E}_{P_{tr}}[\ell(h(z), Y) \mid z]\right|\right]$$

$$\leq \mathcal{O}\left(\sqrt{1 - \exp\left(-s\left(\sup_{Q \in \mathcal{C}(P_{tr})} \mathrm{KL}(Q(Y \mid z) \| P_{tr}(Y \mid z)))\right)\right)}\right). \tag{31}$$

$\square$

**Remark.** The theorem establishes that the generalization gap is upper bounded by the square root of the worst-case KL divergence among training communities, amplified by the expansion function $s(\cdot)$. This means that if predictive mechanisms remain consistent across training communities, the model generalizes better to unseen ones. By explicitly reducing this worst-case divergence, CHASM tightens the bound and enhances robustness against latent community shifts.

## F    LIMITATION AND FUTURE WORK

While CHASM demonstrates strong performance in addressing community-level distribution shifts, several limitations remain. First, the current framework assumes that latent community structures can be sufficiently captured by the mined hardest subgraphs. In practice, communities may overlap or evolve dynamically, which could reduce the effectiveness of static subgraph mining. Second, the adaptive augmentation module relies on simple discrepancy measures (feature statistics and degree distributions), which may not fully capture higher-order structural or semantic variations. Third, CHASM introduces additional computational overhead due to adversarial subgraph extraction, which may limit scalability to extremely large-scale graphs.

For future work, we plan to extend CHASM in several directions. One promising avenue is to incorporate dynamic or hierarchical community modeling, allowing the framework to adapt to overlapping and evolving graph structures. Another direction is to design more expressive discrepancy measures that leverage graph kernels or representation learning techniques to better guide augmentation. Finally, improving the efficiency of adversarial subgraph mining, for example via sampling or scalable approximation algorithms, would further enhance CHASM's applicability to web-scale graph data.

## G    THE USE OF LLMS

We used Grok 4 and GPT 4 as large language models to assist with polishing the English language of the paper, including grammar corrections, phrasing improvements, and clarity enhancements in sections such as the abstract, introduction, and method descriptions. All generated suggestions were reviewed and edited by the authors to ensure accuracy, originality, and alignment with the technical content. The authors take full responsibility for the final version of the paper, including all ideas, analyses, and results.

