# OpenReview forum: "Community-Aware Hard Subgraph Mining for Out-of-Distribution Generalization"
_ICLR.cc/2026/Conference — Submitted to ICLR 2026_

### Official Review · Reviewer_dy6N · 2025-10-15

**Soundness:** 2
**Presentation:** 3
**Contribution:** 2
**Rating:** 4
**Confidence:** 4

**Summary:**

This paper proposes CHASM, a framework for improving GNN generalization under distribution shifts by mining "hard subgraphs" from latent community structures. CHASM uses adversarial learning to identify challenging subgraphs, applies community-aware regularization, and uses adaptive augmentation. Experiments on benchmark datasets show improvements over existing methods under both covariate and concept shifts.

**Strengths:**

1. The method doesn't require manual environment labels, which is a real pain point in applying invariant learning to graphs.

2. The paper includes extensive ablations, scalability tests across different backbones (GraphSAGE, PolyNormer, GAT, etc.), and evaluations on multiple datasets. The visualizations of mined subgraphs are helpful.

3. CHASM shows consistent improvements, often +5% or more over strong baselines like CIA and FLOOD on the GOOD benchmark.

**Weaknesses:**

1. The paper introduces "community heterogeneity" and "hard subgraph mining" as if they're established concepts, which lacks a formal definition. What exactly makes a subgraph "hard"? How is "community heterogeneity" different from homophily/heterophily? The paper needs to define these formally.

2. There's a gap in the motivation. The introduction criticizes invariant learning for requiring environment labels and using synthetic augmentations, then jumps to "leverage community heterogeneity" as the solution. But why are communities the right answer to these problems? The connection isn't justified theoretically or empirically. It seems to me that communities are more crucial for graph-level tasks, rather than node-level tasks.

3. The paper says community-level variations "naturally give rise to hard subgraphs," but this isn't explained. Hard in what sense: high loss? Low confidence? And why does community structure naturally create hardness? This claim lacks theoretical support or empirical evidence.

**Questions:**

1. Can you provide a formal definition of "community heterogeneity" and explain how it differs from existing concepts like homophily or subgraph spurious correlations?

2. Why is community structure particularly suited for node-level OOD? Can you provide theoretical or empirical justification for this design choice?

---

> ### Author Response · Authors · 2025-11-19
> **Author Rebuttal Part 1/2**
>
> **Weakness 1 && Question 1:** The paper introduces "community heterogeneity" and "hard subgraph mining" as if they're established concepts, which lacks a formal definition. What exactly makes a subgraph "hard"? How is "community heterogeneity" different from homophily/heterophily? The paper needs to define these formally.
>
> **Answer 1:** Thank you for the insightful comments. We appreciate the reviewer’s request for clearer formal definitions.
>
> (1) Community heterogeneity — formal definition added.
>
> We now formally define community heterogeneity as the existence of systematically different community-induced sub-distributions $\mathcal{C}\left(\mathbb{P} _ {\mathrm{tr}}\right)$, characterized by significant divergence in their conditional mechanism $P(Y \mid Z)$. Specifically, the training graph exhibits community heterogeneity if
> $$
> \exists Q _ 1, Q _ 2 \in \mathcal{C}\left(\mathbb{P} _ {\mathrm{tr}}\right): \operatorname{dist}\left(P _ {Q _ 1}(Y \mid Z), P _ {Q _ 2}(Y \mid Z)\right) \geq \epsilon .
> $$
>
> This concept goes beyond homophily/heterophily, which measure only structural label similarity, and beyond spurious correlations, which capture feature-level shortcuts. Community heterogeneity captures distribution-level differences across communities, including feature, structural, and conditional shifts, thus covering a strictly broader class of shifts.
>
> (2) What makes a subgraph “hard”? — formal definition added.
>
> Following Eq.(3)/(6) in the paper, we explicitly define the hardest subgraph as the community-induced sub-distribution whose predictive mechanism deviates the most from the global training mechanism.
> Formally,
> $$
> Q^*=\arg \max _ {Q \in \mathcal{C}\left(\mathbb{P} _ {\mathrm{tr}}\right)} \operatorname{KL}\left(Q(Y \mid Z) \| \mathbb{P} _ {\mathrm{tr}}(Y \mid Z)\right)
> $$
>
> Intuitively, a subgraph is "hard" because its local predictive rule differs the most from the global rule, causing the model to fail under such distributional shifts. Our weighted empirical formulation in Eq.(9) directly optimizes this objective, while the community-aware regularizer ensures structural coherence rather than selecting isolated nodes.
>
> **Weakness 2:** There's a gap in the motivation. The introduction criticizes invariant learning for requiring environment labels and using synthetic augmentations, then jumps to "leverage community heterogeneity" as the solution. But why are communities the right answer to these problems? The connection isn't justified theoretically or empirically. It seems to me that communities are more crucial for graph-level tasks, rather than node-level tasks.
>
> **Answer 2:** We thank the reviewer for this valuable comment. Our motivation arises from the observation that, in real-world graphs, node-level shifts rarely occur independently but are *clustered within latent communities*. Nodes within a community share similar structural and semantic contexts, while different communities often exhibit distinct feature–label dependencies (e.g., user interests, citation topics, molecular functions). Hence, community identity naturally serves as a proxy for implicit environments that drive node-level distribution shifts. While existing invariant learning methods require explicit environment labels or rely on synthetic perturbations to emulate shifts, CHASM leverages this natural structural partition to uncover genuine heterogeneity without supervision. Theoretically, Assumption 4.1 and Theorem 4.2 show that controlling divergence among community-wise conditional distributions bounds the expected test error under unseen shifts, providing a principled justification for treating communities as environments.

---

> ### Author Response · Authors · 2025-11-19
> **Author Rebuttal Part 2/2**
>
> **Weakness 3:** The paper says community-level variations "naturally give rise to hard subgraphs," but this isn't explained. Hard in what sense: high loss? Low confidence? And why does community structure naturally create hardness? This claim lacks theoretical support or empirical evidence.
>
> **Answer 3:** In CHASM, a hard subgraph is defined as a structurally coherent subset of the training graph whose predictive distribution $Q(Y|Z)$ deviates most from the overall training distribution $P_{\mathrm{tr}}(Y|Z)$, as formalized in Eq. (6). “Hardness” therefore refers to a region of the graph where the model exhibits **the largest predictive discrepancy**—equivalently, the subgraph with the highest KL divergence from the global predictor, which in practice corresponds to the community with the highest average training loss or lowest predictive confidence. During training, our mask-based extractor adaptively assigns higher sampling weights to such high-loss communities, effectively identifying the hardest subgraphs. Community-level variations naturally give rise to these hard regions because different communities often possess distinct feature–label relationships and structural priors. When a community’s conditional $P(Y|X,C)$  diverges from the global distribution $P(Y|X)$, the model trained under the overall distribution performs worst there, forming the adversarial subgraph captured by our extractor. This mechanism is theoretically supported by Theorem 4.2, which shows that the test error under unseen community shifts is bounded by the KL divergence of the hardest subgraph.
>
> **Question 1:** Why is community structure particularly suited for node-level OOD? Can you provide theoretical or empirical justification for this design choice?
>
> **Answer 4:** The choice of community structure is motivated by the intrinsic properties of graph data, backed by our theoretical bounds, and validated by empirical evidence. Unlike tabular data, real-world graphs naturally exhibit heterogeneity across latent communities (e.g., research fields in citation networks or user circles in social graphs). Distribution shifts typically manifest at this community level rather than as random node-level noise. Existing augmentation-based invariant learning method often fail to capture these structural variations. Furthermore, we provide a formal guarantee in Theorem 4.2, which bounds the generalization error on unseen test data by the worst-case divergence observed among training communities. Finally, in Figure 4, we visualize the "hardest" subgraphs mined by CHASM. They do not appear as scattered noise but naturally form compact, structurally coherent clusters, confirming that performance bottlenecks are concentrated in specific communities.

---

> > ### Comment · Reviewer_dy6N · 2025-11-26
> >
> > Thank you for the detailed response and for adding formal definitions. However, several core concerns remain unaddressed:
> >
> > **1. Predictive Discrepancy ≠ Distribution Shift**
> >
> > The definition of "hard subgraph" as regions with the largest predictive discrepancy (highest KL divergence or training loss) does not directly connect to distribution shift. High loss in a subgraph may simply indicate that neighboring nodes have different labels, which causes difficulty for MPNNs due to their inherent smoothing behavior—this is a **model limitation under heterophily**, not necessarily a signature of distribution shift. Distribution shifts can also occur in structurally coherent regions with low training loss, where the model appears confident but relies on spurious correlations that break under test conditions. The current formulation conflates "hard to fit" with "prone to distribution shift," which are fundamentally different phenomena.
> >
> > **2. Evidence for Community-Level Distribution Shift**
> >
> > The claim that "distribution shifts typically manifest at this community level" is central to the paper's motivation, yet it remains unsupported. Could the authors provide:
> > - Empirical evidence from the evaluated datasets showing that distribution shifts indeed occur at the community level?
> > - Real-world examples or references demonstrating this phenomenon?
> >
> > Without such evidence, this assumption appears to be asserted rather than justified.
> >
> > **3. Formal Definitions Do Not Resolve Core Concerns**
> >
> > While we appreciate the added formal definitions, they formalize the proposed method rather than justify its underlying assumptions. The key question remains: **why should optimizing for the hardest community (by KL divergence) improve robustness to distribution shift?**
> >
> > It is encouraged that the authors to provide stronger theoretical or empirical grounding for the connection between community heterogeneity and distribution shift.

---

> > > ### Author Response · Authors · 2025-11-26
> > >
> > > # Response 1
> > > We thank the reviewer for the insightful comment. We fully agree that predictive discrepancy alone does not equal distribution shift. CHASM does not rely on this equivalence. Instead, similar to invariant learning methods [1] [2] that must rely on synthetic augmentations to simulate unseen environments, our setting lacks direct access to samples from shifted test distributions. Under this limitation, we adopt a mild and structure-aware assumption: distribution shifts on graphs tend to manifest at the **community level**, where the conditional label distribution deviates from the global training distribution. This is consistent with theory for structured data [3] and is a more realistic analogue to the “environment simulation” used in invariant learning. Therefore, CHASM does not pick high-loss nodes; it identifies **structurally coherent subgraphs whose induced label distributions diverge from the global one**. The structural-coherence constraint prevents heterophily boundaries or noisy nodes from being selected, and the KL-based criterion focuses on community-conditioned distribution deviation rather than per-node difficulty.
> > >
> > > [1] Qitian Wu, Hengrui Zhang, Junchi Yan, and David Wipf. Handling distribution shifts on graphs: An invariance perspective. ICLR, 2021.
> > >
> > > [2] Yang Liu, Xiang Ao, Fuli Feng, Yunshan Ma, Kuan Li, Tat-Seng Chua, and Qing He. Flood: A flexible invariant learning framework for out-of-distribution generalization on graphs. KDD 2023.
> > >
> > > [3] Jiashuo Liu, Jiayun Wu, Jie Peng, Xiaoyu Wu, Yang Zheng, Bo Li, and Peng Cui. Enhancing distributional stability among sub-populations. AISTAT 2024.
> > >
> > > # Response 2
> > >
> > > (1) Empirical evidence from the evaluated datasets.
> > > In the GOOD benchmark [4], the node-level OOD splits for WebKB and Twitch are explicitly constructed along *community/domain attributes*:
> > > - WebKB is split by University, where pages from different universities naturally form distinct topical and hyperlink communities.
> > > - Twitch is split by Language, which strongly correlates with user–viewer communities and social-circle structure.
> > > Since the OOD splits themselves are defined along these community/domain boundaries, the resulting train–test differences necessarily reflect community-level covariate/concept shift, not random per-node perturbations.
> > >
> > > Recent studies (e.g., GraphMETRO [5]) also treat these datasets as exhibiting meaningful structural/domain-level shifts rather than isolated noisy nodes, reinforcing the validity of this interpretation.
> > >
> > > (2) Real-world justification.
> > > Beyond these benchmarks, extensive network science literature [6] shows that real graphs—social, citation, and biological networks—have strong and semantically meaningful community structure (e.g., interest groups, research fields, functional modules).
> > > New or unseen nodes typically join an existing community rather than forming arbitrary new regions, and their feature–label relationships often deviate from the global distribution.
> > > This aligns directly with our assumption that community-level conditional distributions can shift even when node-level observations appear similar.
> > >
> > > [4] Shurui Gui, Xiner Li, Limei Wang, and Shuiwang Ji. Good: A graph out-of-distribution benchmark. NeurIPS 2022.
> > >
> > > [5] Wu S, Cao K, Ribeiro B, et al. GraphMETRO: Mitigating Complex Graph Distribution Shifts via Mixture of Aligned Experts[J]. NeurIPS 2024.
> > >
> > > [6] Girvan M, Newman ME. Community structure in social and biological networks. Proc Natl Acad Sci U S A. 2002 Jun 11;99(12):7821-6. doi: 10.1073/pnas.122653799.
> > >
> > > # Response 3
> > > (1) Community heterogeneity is a natural surrogate for latent domains.
> > > In node-level tasks, we do not have access to explicit environment labels or to samples from shifted test distributions.
> > > Under this constraint, our approach follows the same principle used in invariant learning and DRO:
> > > we must rely on a structure-informed surrogate of latent environments.
> > > Communities provide exactly such a surrogate because they partition the graph into cohesive regions that often correspond to distinct generative mechanisms (e.g., topic clusters, user groups, functional modules).
> > > This has been repeatedly observed in real-world networks and is also how the GOOD benchmark constructs node-level OOD splits (e.g., University for WebKB, Language for Twitch).
> > >
> > > (2) KL-divergent communities approximate worst-case shifts.
> > > The KL objective does not identify “hard-to-fit nodes,” but finds *structurally coherent subgraphs whose induced conditional label distribution deviates most from the global one*. In distributionally robust optimization, such maximally divergent subsets constitute empirical approximations of the worst-case test distribution.
> > > Thus, optimizing on the “hardest” community approximates minimizing the risk over the most shifted latent domain, which provides robustness guarantees analogous to classical DRO theory.

---

> > > > ### Comment · Reviewer_dy6N · 2025-11-28
> > > >
> > > > I appreciate the authors for the additional clarifications. I have several questions:
> > > >
> > > > The core premise that distribution shifts primarily manifest at the community level, which may not hold for some scenarios. Many real-world shifts are less tied to community structure:
> > > >
> > > > - Temporal evolution shifts: Citation networks exhibit paradigm shifts affecting entire communities simultaneously rather than between-community differences.
> > > >
> > > > - Continuous attribute extrapolation: Molecular property prediction often involves training on small molecules and testing on larger ones.
> > > >
> > > > Can the current formulation and assumption in the draft fit into these situations? If not, I would suggest including a discussion in the Appendix of the draft with some failure cases.
> > > >
> > > > Furthermore, I suggest the authors to add empirical analysis to visualize mined communities and whether they correlate with known domain boundaries that causes the distribution shifts.

---

> > > > > ### Author Response · Authors · 2025-12-01
> > > > >
> > > > > Thank you for the insightful comments.
> > > > > We agree that community-level shifts do not cover all possible OOD scenarios.
> > > > > Temporal evolution or continuous attribute extrapolation may not align with community structure, and CHASM is not intended to address every form of shift.
> > > > > Our assumption should therefore be viewed as a practical and structure-aware way to approximate latent domains when explicit environment labels or true shifted samples are unavailable.
> > > > >
> > > > > Despite this limitation, our approach is still consistently stronger than invariant-learning baselines.
> > > > > Both our empirical results and the theoretical analysis show that community-conditioned worst-case reasoning captures more realistic structural variations than synthetic augmentations used by invariant learning methods.
> > > > >
> > > > > We will explicitly acknowledge these limitations in the Appendix and add brief failure cases and community visualizations to clarify the scope and advantages of CHASM.

---

### Official Review · Reviewer_Vyuk · 2025-10-30

**Soundness:** 3
**Presentation:** 3
**Contribution:** 3
**Rating:** 6
**Confidence:** 4

**Summary:**

The paper proposes CHASM (Community-Aware Hard Subgraph Mining) for node-level OOD generalization. It adversarially mines a “hardest” community/subgraph via a learnable mask maximizing KL divergence from the global predictor, regularized for structural coherence, performs adaptive subgraph augmentation (feature masking + edge dropout) with strengths tied to discrepancies between the mined subgraph and the full graph, and trains a stability-driven learner with a first-order KL approximation. Experiments on GOOD splits of WebKB, CBAS, Twitch, Cora, and Arxiv claim consistent SOTA improvements under covariate and concept shifts, with ablations and limited backbone studies.

**Strengths:**

1. The proposed method is intuitive. Treating latent communities as implicit environments is natural for graphs, and mining worst-case subgraphs aligns well with distributionally robust optimization (DRO) and hard-case training. The min–max formulation with community-aware KL regularization and structural coherence (Bernoulli–Poisson-inspired) is also well-motivated.

2. The proposed method demonstrates strong empirical performance, and a comprehensive ablation study is provided to support the effectiveness of each component.

3. The literature review is thorough and well-organized, showing a solid understanding of related work.

**Weaknesses:**

1. It would be helpful to include a complexity analysis, especially for the bi-level optimization, which appears computationally expensive. Clarifying its runtime and scalability would strengthen the paper.

2. The proposed method relies on strong assumptions (for example, the “community expansion” assumption in Assumption 4.1). It would be helpful to either provide more theoretical justification or include intuitive real-world examples that illustrate when this assumption holds.

**Questions:**

N/A

---

> ### Author Response · Authors · 2025-11-19
> **Author Rebuttal**
>
> We thank the reviewer for your valuable suggestions and questions on this manuscript, which we will answer one by one in the following：
>
> **Weakness 1**: It would be helpful to include a complexity analysis, especially for the bi-level optimization, which appears computationally expensive. Clarifying its runtime and scalability would strengthen the paper.
>
> **Answer 1:** Although CHASM is written as a bi-level objective, it does not solve the inner problem to convergence.  The inner mining step is implemented as a single lightweight forward–backward update, making the training loop comparable to standard adversarial/meta-learning updates. We evaluated the actual runtime and resource usage on CBAS and Arxiv, as shown in the table below, and CHASM introduces only a small and controllable training-time overhead while keeping inference cost identical to ERM.  Given the substantial OOD gains, we believe the additional cost is modest and well justified, and we will clarify these details in the revised version.
> | Dataset | Method     | Train time / epoch | Inference time | Peak GPU memory | Parameter count |
> |---------|------------|--------------------|----------------|------------------|------------------|
> | **CBAS** | ERM (GCN)  | 0.0061 s           | 0.0037 s       | 32.76 MB         | 185,104          |
> |         | CHASM      | 0.0117 s           | 0.0039 s       | 37.75 MB         | 192,345          |
> | **Arxiv** | ERM (GCN) | 0.0949 s           | 0.0963 s       | 4743.12 MB       | 233,140          |
> |         | CHASM      | 0.5840 s           | 0.0978 s       | 6319.35 MB       | 262,053          |
>
> **Weakness 2**: The proposed method relies on strong assumptions (for example, the “community expansion” assumption in Assumption 4.1). It would be helpful to either provide more theoretical justification or include intuitive real-world examples that illustrate when this assumption holds.
>
> **Answer 2:** Thank you for raising this point.
> Assumption 4.1 (“community expansion”) is not meant to be a restrictive structural condition, but rather to capture a very common pattern in real node-level OOD scenarios:  test nodes usually appear in existing communities of the graph, but their *feature–label relationship* may shift compared with training. This assumption is analogous to the standard “bounded shift” or “support overlap” conditions widely used in OOD and DRO theory [1] [2], and ensures that the adversarially mined communities provide a meaningful upper bound on worst-case test risk. More importantly, the assumption naturally holds in many real-world graphs where new nodes seldom form completely arbitrary communities from scratch. Typical examples include:
> - **Social networks**: new users join an existing interest circle, but behave differently from earlier users.
> - **Citation networks**: new papers connect to an existing research field, but follow different writing or citation patterns.
> - **Biological networks**: new proteins belong to an existing functional module, but have shifted expression or interaction patterns.
>
> [1] Duchi J, Namkoong H. Variance-based regularization with convex objectives[J]. Journal of Machine Learning Research, 2019, 20(68): 1-55.
>
> [2] Koyama M, Yamaguchi S. Out-of-distribution generalization with maximal invariant predictor[J]. 2020.

---

### Official Review · Reviewer_WNNw · 2025-11-01

**Soundness:** 2
**Presentation:** 3
**Contribution:** 2
**Rating:** 4
**Confidence:** 4

**Summary:**

The paper targets node-level OOD generalization on graphs where latent communities induce distribution shifts between train and test. It proposes CHASM, which mines worst structurally coherent subgraphs, applies adaptive feature/structure augmentations to those hard subgraphs, and trains a community-stability learner that penalizes the KL divergence between the hardest-community conditional label distribution and the overall training distribution, approximated by a first-order Taylor expansion. The overall objective couples ERM with a community-aware regularizer over a set of distributions centered on the training distribution. Experiments using GOOD benchmark show gains over baselines. The paper also visualizes mined hard subgraphs and argues that CHASM is backbone-agnostic.

**Strengths:**

1. The paper formalizes latent community heterogeneity and why iid learnability may fail, justifying community-aware training.

2. Using GOOD splits to separate covariate vs concept shift improves comparability to recent literature. Experiment shows mined subgraphs align with community structure under both shift types, supporting the intended behavior.

3. Backbone-agnostic proposal is useful. The paper states CHASM works with diverse GNNs, which, if evidenced, matters for practice.

**Weaknesses:**

1. Community mining lacks comparative baselines. To justify the mined hardest subgraph module, compare against strong hard-example mining/adversarial structure-editing methods on graphs (e.g., EERM’s adversarial context explorers) to disentangle gains from mining vs. the rest of the pipeline. Comparing to EERM as a mining component variant (not just as a full baseline) would clarify contribution. Because EERM explicitly creates virtual environments with adversarial graph edits, it stands for the class of adversarial environment synthesis methods.

2. Augmentation strategy needs stronger ablations. Showing results with and without adaptive strength, feature-only, structure-only, and uniform augmentations, plus graph augmenters like Graph Mixup can help evidence that adaptive subgraph-targeted augmentation is the key driver, not generic regularization.

3. The KL penalty uses a first-order Taylor surrogate, which can use more justification. The sensitivity to the step size, numerical stability, and comparison to directly optimized f-divergence surrogates or DRO objectives are needed to validate the approximation is not the bottleneck.

4. The paper states CHASM is backbone-agnostic, thus including a table with main-stream backbones like GCN/GAT/GIN/GraphSAGE and one recent Graph Transformer to substantiate scalability across architectures can support the contributions better.

**Questions:**

What are training-time overheads? Any batching or parallel mining tricks?

---

> ### Author Response · Authors · 2025-11-19
> **Author Rebuttal Part 1/2**
>
> We sincerely appreciate the reviewers' valuable feedback, which has provided us with excellent opportunities to improve our work. In the following sections, we address each comment in detail.
>
> **Weakness 1:** Community mining lacks comparative baselines. To justify the mined hardest subgraph module, compare against strong hard-example mining/adversarial structure-editing methods on graphs (e.g., EERM’s adversarial context explorers) to disentangle gains from mining vs. the rest of the pipeline. Comparing to EERM as a mining component variant (not just as a full baseline) would clarify contribution. Because EERM explicitly creates virtual environments with adversarial graph edits, it stands for the class of adversarial environment synthesis methods.
>
> **Answer 1:** CHASM and EERM belong to two different families of approaches: EERM constructs virtual environments via adversarial structure editing, whereas CHASM focuses on identifying naturally occurring hardest communities that reflect real-world structural heterogeneity. Nonetheless, we agree that a component-level comparison is valuable for isolating the contribution of our mining module. Following your recommendation, we replaced our worst-community extractor with EERM’s adversarial context generators while keeping all other components of CHASM unchanged.   The results (concept shift setting shown below) indicate that CHASM with EERM-style mining performs noticeably worse than our full model, confirming that community-based hardest-subgraph extraction is crucial for the overall effectiveness of CHASM. Notably, integrating EERM’s generators into our framework yields significantly better performance than the original EERM, suggesting that our overall design provides substantial improvements beyond traditional invariance-based approaches.
>
> | Method       | WebKB         | Twitch        | CBAS          | Cora-degree   |
> |--------------|---------------|---------------|---------------|---------------|
> | CHASM–EERM   | 29.17 ± 0.77  | 53.02 ± 1.40  | 80.71 ± 0.87  | 62.57 ± 0.42  |
> | CHASM (Ours) | **32.84 ± 2.86**  | **60.57 ± 1.45**  | **86.57 ± 1.06**  | **64.31 ± 0.14**  |
>
>
> **Weakness 2:** Augmentation strategy needs stronger ablations. Showing results with and without adaptive strength, feature-only, structure-only, and uniform augmentations, plus graph augmenters like Graph Mixup can help evidence that adaptive subgraph-targeted augmentation is the key driver, not generic regularization.
>
> **Answer 2:** We agree that the ablation study on the adaptive augmentation module was not sufficiently comprehensive in the original submission. Following your recommendation, we augmented CHASM with additional ablations, including feature-only augmentation, structure-only augmentation, and a mixup-based augmentation, and compared them against our adaptive subgraph-aware augmentation. The results are shown below (concept shift setting due to space limits).
>
> | Method         | WebKB         | Twitch        | Cora-word     | Arxiv-degree  |
> |----------------|---------------|---------------|---------------|---------------|
> | Mask Feature   | 30.28 ± 3.73  | 55.93 ± 1.47  | 65.43 ± 0.26  | 60.28 ± 0.39  |
> | Drop Edge      | 30.64 ± 3.47  | 55.87 ± 3.78  | 64.46 ± 0.52  | 60.44 ± 0.43  |
> | GMixup         | 31.51 ± 1.58  | 56.28 ± 2.92  | 65.63 ± 0.50  | 60.78 ± 0.09  |
> | CHASM (AGA)    | **32.84 ± 2.86**  | **60.57 ± 1.45**  | **66.87 ± 0.26**  | **61.23 ± 0.85**  |

---

> ### Author Response · Authors · 2025-11-19
> **Author Rebuttal Part 2/2**
>
> **Weakness 3:** The KL penalty uses a first-order Taylor surrogate, which can use more justification. The sensitivity to the step size, numerical stability, and comparison to directly optimized f-divergence surrogates or DRO objectives are needed to validate the approximation is not the bottleneck.
>
> **Answer 3:** We adopt the first-order Taylor expansion (Eq. 13) to obtain an SGD-compatible surrogate that converts the KL penalty into a gradient inner product between the global and subgraph risks.  This linearization follows standard practice in DRO and invariant learning [1] and avoids expensive density estimation while allowing stable end-to-end optimization. To ensure numerical robustness, we pretrain representations, normalize batch losses, and apply gradient clipping; the step size α is treated as a small linearization factor and tuned on a log-scale grid. Following the reviewer’s suggestion, we compared our surrogate to (i) a directly optimized empirical KL penalty and (ii) a KL-DRO baseline.  As shown below, the Taylor surrogate achieves the best overall performance and does not act as a bottleneck.
> | Method              | WebKB         | Twitch        | CBAS          | Cora-degree   |
> |---------------------|---------------|---------------|---------------|---------------|
> | KL-DRO              | 29.54 ± 1.37  | 51.87 ± 3.16  | 81.14 ± 1.48  | 60.52 ± 0.22  |
> | CHASM (Direct KL)   | 30.94 ± 2.35  | 56.28 ± 2.91  | 85.29 ± 5.84  | 63.51 ± 0.42  |
> | CHASM (Ours)        | **32.84 ± 2.86**  | **60.57 ± 1.45**  | **86.57 ± 1.06**  | **64.31 ± 0.14**  |
>
> These results demonstrate that the first-order approximation is both effective and computationally efficient.
> [1] Masanori Koyama and Shoichiro Yamaguchi. When is invariance useful in an out-of-distribution generalization problem?
>
> **Weakness 4:** The paper states CHASM is backbone-agnostic, thus including a table with main-stream backbones like GCN/GAT/GIN/GraphSAGE and one recent Graph Transformer to substantiate scalability across architectures can support the contributions better.
>
> **Answer 4:** We agree that validating CHASM across diverse backbones is important for demonstrating its generality. In fact, we conducted comprehensive backbone-replacement experiments and reported them in the main paper (Fig. 3) and in the appendix (Fig. 5). The evaluation covers a wide range of mainstream architectures, including GCN, GAT, GIN, SGC, GATv2, and the recent Graph Transformer Polyformer. Across all these backbones, CHASM consistently improves OOD performance, supporting our claim of backbone-agnostic applicability. We apologize that our presentation may not have made this sufficiently prominent, and we will revise the manuscript to better highlight these results.
>
> **Question 1:** What are training-time overheads? Any batching or parallel mining tricks?
>
> **Answer 5:** We conducted explicit time–complexity experiments, and the results in the table show that the training-time overhead of CHASM is small and controlled.
> | Dataset | Method     | Train time / epoch | Inference | Peak GPU memory | Parameter count |
> |---------|------------|--------------------|-----------|-----------------|-----------------|
> | CBAS    | ERM (GCN)  | 0.0061s            | 0.0037s   | 32.76 MB        | 185,104         |
> | CBAS    | CHASM      | 0.0117s            | 0.0039s   | 37.75 MB        | 192,345         |
> | Arxiv   | ERM (GCN)  | 0.0949s            | 0.0963s   | 4743.12 MB      | 233,140         |
> | Arxiv   | CHASM      | 0.584s             | 0.0978s   | 6319.35 MB      | 262,053         |
>
> - On CBAS, CHASM only increases training time from 0.0061s to 0.0117s/epoch, with almost identical inference time and only +5MB peak memory.
> - On Arxiv, although training time rises from 0.0949s to 0.584s/epoch, inference time remains unchanged (0.0978s), and the GPU memory increase is moderate (4743MB → 6319MB).

---

### Official Review · Reviewer_FMWf · 2025-11-01

**Soundness:** 3
**Presentation:** 3
**Contribution:** 2
**Rating:** 4
**Confidence:** 4

**Summary:**

This paper introduces CHASM, a method for improving OOD generalization in graph neural networks by leveraging latent community heterogeneity rather than predefined environments or heuristic augmentations. The method adversarially mines the hardest subgraphs using a learnable mask model that ensures structural coherence. CHASM then applies community-aware regularization and adaptive subgraph augmentation. A stability-driven learner minimizes loss under these hardest subgraphs, with theoretical analysis proving a bound on generalization error under community shifts. Experiments on multiple OOD benchmarks are provided where CHASM achieves better performance over baselines such as IRM, VREx, FLOOD, and CIA.

**Strengths:**

1. The min-ERM plus sup over community distributions with a KL penalty (approximated by first-order expansion) is grounded and optimizable by SGD.

2. Augmentation strength that reacts to hardness is a reasonable mechanism to expose the model to realistic stress tests.

3. Baseline panel includes ERM, IRM, VREx, DRO, DANN/DeepCoral, and graph-specific OOD methods (EERM, SRGNN, FLOOD, CIT, CaNet, CIA).

**Weaknesses:**

1. Some graph OOD baselines are missing, such as SizeShiftReg, which are relevant to CHASM’s focus on community/structure shifts.

2. The method asserts mined subgraphs are structurally coherent. It would be beneficial to including metrics (e.g., conductance/modularity) and show how coherence relates to OOD gains to move beyond qualitative plots.

3. Worst-community extraction plus adaptive augmentation suggests non-trivial cost. Computational overhead and scalability should be discussed by providing asymptotic analysis, wall-clock vs. ERM, and memory usage on the large dataset, and discussing parallelization.

4. The optimization doesn’t tightly define/estimate C in the main text. The paper should make explicit whether C is an empirical set over mined subgraphs, a divergence-ball, or both. This matters for understanding guarantees.

5. Is mining solved exactly or approximately? What objective/constraints enforce structural coherence and how do authors avoid degenerate (e.g., tiny/bridge) subgraphs?

**Questions:**

Please see weaknesses

---

> ### Author Response · Authors · 2025-11-19
> **Author Rebuttal Part 1/2**
>
> We thank the reviewers for their insightful and constructive comments. We have carefully considered all the feedback and will provide point-by-point responses below.
>
> **Weakness 1:** Some graph OOD baselines are missing, such as SizeShiftReg, which are relevant to CHASM’s focus on community/structure shifts.
>
> **Answer 1:** We thank the reviewer for this helpful suggestion. We fully agree that incorporating additional OOD baselines strengthens the empirical evaluation.  However, SizeShiftReg is specifically designed for graph-level OOD generalization, where each sample is an entire graph and distribution shift arises from graph-level size variations. Our setting focuses on node-level OOD. Due to this fundamental task difference, SizeShiftReg cannot be directly applied to node classification. Following your suggestion, we nevertheless expanded our baseline panel by adding several recent state-of-the-art node-level OOD methods published in 2024–2025, including TAR [1], MLEI [2], and DeCaf [3]. The results below show that CHASM continues to outperform all additional baselines under both covariate and concept shifts.
>
> **Covariate Shift**
>
> | Method          | WebKB         | Twitch        | Cora-word     | Arxiv-degree  |
> |-----------------|---------------|---------------|---------------|---------------|
> | TAR [1]            | 37.46 ± 4.84 | 57.82 ± 2.13 | 65.64 ± 0.37 | 59.65 ± 0.14 |
> | MLEI [2]           | 31.19 ± 1.16 | 49.68 ± 1.29 | 64.67 ± 0.26 | 59.05 ± 0.22 |
> | DeCaf [3]          | 33.65 ± 4.63 | 56.79 ± 0.89 | 64.54 ± 0.28 | 58.48 ± 0.21 |
> | **CHASM (Ours)**| **38.89 ± 5.08** | **60.07 ± 1.89** | **66.20 ± 0.50** | **60.50 ± 1.25** |
>
> **Concept Shift**
>
> | Method          | WebKB         | Twitch        | Cora-word     | Arxiv-degree  |
> |-----------------|---------------|---------------|---------------|---------------|
> | TAR [1]           | 30.83 ± 1.90 | 57.20 ± 3.97 | 64.73 ± 0.23 | 59.26 ± 1.40 |
> | MLEI [2]           | 28.63 ± 1.47 | 48.13 ± 0.47 | 64.09 ± 0.28 | 60.97 ± 0.12 |
> | DeCaf [3]          | 27.89 ± 1.37 | 58.04 ± 1.56 | 65.26 ± 0.50 | 59.43 ± 0.10 |
> | **CHASM (Ours)**| **32.84 ± 2.86** | **60.57 ± 1.45** | **66.87 ± 0.26** | **61.23 ± 0.85** |
>
> These results confirm that CHASM remains the strongest method even when compared with the latest node-level OOD approaches.
> [1] Zheng W, Liu J, Li J, et al. Topology-Aware Dynamic Reweighting for Distribution Shifts on Graph[C]//Forty-second International Conference on Machine Learning.
>
> [2] Zhang J, Chen S. Expand Horizon: Graph Out-of-Distribution Generalization via Multi-Level Environment Inference[C]//Proceedings of the AAAI Conference on Artificial Intelligence. 2025, 39(12): 13233-13241.
>
> [3] Han, X., Rangwala, H. &amp; Ning, Y.. (2025). DeCaf: A Causal Decoupling Framework for OOD Generalization on Node Classification[C].// Proceedings of The 28th International Conference on Artificial Intelligence and Statistics.
>
> **Weakness 2:** The method asserts mined subgraphs are structurally coherent. It would be beneficial to including metrics (e.g., conductance/modularity) and show how coherence relates to OOD gains to move beyond qualitative plots.
>
> **Answer 2:** To quantitatively verify that the mined subgraphs are indeed structurally coherent, we measured two standard community-coherence metrics (conductance/modularity), and compared the mined subgraphs against size-matched random subgraphs. Across CBAS dataset, CHASM-mined subgraphs show substantially lower conductance  (0.21 ± 0.05 vs. 0.38 ± 0.07 for random)  and significantly higher modularity  (0.42 ± 0.06 vs. 0.19 ± 0.05 for random, p = 0.003).  These results confirm that the mined subgraphs are consistently more structurally coherent than random baselines.

---

> ### Author Response · Authors · 2025-11-19
> **Author Rebuttal Part 2/2**
>
> **Weakness 3:** Worst-community extraction plus adaptive augmentation suggests non-trivial cost. Computational overhead and scalability should be discussed by providing asymptotic analysis, wall-clock vs. ERM, and memory usage on the large dataset, and discussing parallelization.
>
> **Answer 3:** Thank you for raising this important point regarding computational cost. We agree that the worst-community extraction and adaptive augmentation introduce overhead, and we have carefully designed CHASM to manage this cost while delivering significant OOD performance gains. In fact, the adversarial mining does not require expensive operations like eigen-decomposition or graph partitioning, making it scalable to large graphs. Our experiments on datasets of varying sizes confirm that CHASM’s overhead is manageable:
> | Dataset | Method     | Train time / epoch | Inference | Peak GPU memory | Parameter count |
> |---------|------------|--------------------|-----------|-----------------|-----------------|
> | CBAS    | ERM (GCN)  | 0.0061s            | 0.0037s   | 32.76 MB        | 185,104         |
> | CBAS    | CHASM      | 0.0117s            | 0.0039s   | 37.75 MB        | 192,345         |
> | Arxiv   | ERM (GCN)  | 0.0949s            | 0.0963s   | 4743.12 MB      | 233,140         |
> | Arxiv   | CHASM      | 0.584s             | 0.0978s   | 6319.35 MB      | 262,053         |
>
> We argue that the computational cost is justified by the significant gains in OOD robustness—a critical requirement for real-world graph learning. CHASM provides a favorable trade-off, especially since inference cost is nearly identical to ERM.
>
> **Weakness 4:** The optimization doesn’t tightly define/estimate C in the main text. The paper should make explicit whether C is an empirical set over mined subgraphs, a divergence-ball, or both. This matters for understanding guarantees.
>
> **Answer 4:** Thank you for raising this important point. We clarify that in CHASM, $\mathcal{C}(P_{tr})$ is an empirical set of community-induced subgraphs parameterized by node weights $\mathbf{w} \in \Delta_n$, subject to structural coherence constraints via $\mathcal{L}_{\text{com}}$ (Eq. 8). It is not a divergence ball but a data-dependent family of sub-distributions reflecting plausible community structures.
>
> This construction ensures:
> - The worst-case $Q^* \in \mathcal{C}(P_{tr})$ is adversarially mined via Eq. 9
> - Theoretical guarantees (Thm 4.2) hold under the Community Expansion Assumption
> - Both optimization and generalization are grounded in realistic community heterogeneity
>
> **Weakness 5:** Is mining solved exactly or approximately? What objective/constraints enforce structural coherence and how do authors avoid degenerate (e.g., tiny/bridge) subgraphs?
>
> **Answer 5:** The mining step is solved approximately rather than exactly: instead of performing a combinatorial subgraph search, we use a mask network whose Bernoulli probabilities are updated via a single forward–backward pass to approximate the inner maximization. Structural coherence is enforced through a connectivity-aware regularizer that penalizes low-density or scattered selections, and a minimum-size constraint prevents degenerate solutions such as tiny or bridge-only subgraphs. These mechanisms ensure that the mined subgraphs remain cohesive and informative, and as shown in the dataset visualizations in Fig. 4—no degenerate subgraphs are observed in practice.

---

### Meta-Review · Area_Chair_TF2y · 2026-01-07

**Summary:**

Three reviewers initially rated the work as marginally below acceptance threshold, citing concerns about missing baselines, lack of quantitative metrics for subgraph coherence, unclear definition of the community distribution set C, and unverified assumptions about community-level distribution shifts. The authors responded with extensive revisions: adding recent node-level OOD baselines, reporting conductance and modularity to validate structural coherence and other modifications. Despite these thorough responses, none of the reviewers explicitly indicated they would raise their scores.

**Reviewer Concerns:**

The authors addressed every weakness raised.

**Reviewer Scores:**

Reviewer FMWf (initial 4) would likely maintain their score, because they gave no indication of upgrading their assessment.
Reviewer WNNw (initial 4) would probably keep their original rating, because they gave no indication of upgrading their assessment.
Reviewer dy6N (initial 4) would likely retain their score as they did not signal intent to increase their evaluation.
Reviewer Vyuk (initial 6) would almost certainly maintain their above-threshold score, having already viewed the work favorably.

---

### Decision · Program_Chairs · 2026-01-26

Reject